# Photothermal and Hydrophobic Surfaces with Nano-Micro Structure: Fabrication and Their Anti-Icing Properties

**DOI:** 10.3390/nano15050378

**Published:** 2025-02-28

**Authors:** Meng Li, Renping Ma, Chaokun Yang, Lebin Wang, Shuangqi Lv, Xin Zhao, Mengyao Pan, Jianjian Zhu, Hongbo Xu

**Affiliations:** 1College of Aviation Engineering, Civil Aviation Flight University of China, Guanghan 618307, China; yangchaokun@cafuc.edu.cn (C.Y.); lyushq@cafuc.edn.cn (S.L.); zhaox@cafuc.edu.cn (X.Z.); zhujj.work@cafuc.edu.cn (J.Z.); 2Beijing Vocational College of Labour and Social Security, Beijing 102200, China; 2016010279@bvclss.edu.cn; 3Instrumental Analysis & Research Center, Sun Yat-sen University, Guangzhou 510000, China; wanglb33@mail.sysu.edu.cn; 4Institute of Fundamental and Frontier Sciences, University of Electronic Science and Technology, Chengdu 611731, China; 5School of Chemistry and Chemical Engineering, Harbin Institute of Technology, Harbin 150001, China

**Keywords:** photothermal and hydrophobic surface, nano-micro structure, anti-icing, fabrication methods

## Abstract

The formation of ice due to global climate change poses challenges across multiple industries. Traditional anti-icing technologies often suffer from low efficiency, high energy consumption, and environmental pollution. Photothermal and hydrophobic surfaces with nano-micro structures (PHS-NMSs) offer innovative solutions to these challenges due to their exceptional optical absorption, heat conversion capabilities, and unique surface water hydrophobic characteristics. This paper reviews the research progress of PHS-NMSs in their anti-icing applications. It introduces the mechanisms of ice prevention, fabrication methods, and pathways for performance optimization of PHS-NMSs. The anti-icing performance of PHS-NMSs in different application scenarios is also discussed. Additionally, the paper provides insights into the challenges and future development directions in this field.

## 1. Introduction

With the intensification of global climate change, the surface icing of transportation vehicles such as cars, trains, and aircraft has become increasingly severe under extreme weather conditions, posing significant challenges to traffic safety and efficiency. Icing can alter the aerodynamic properties of vehicles, increase their weight, and reduce mechanical efficiency, potentially leading to structural damage [1]. For instance, icing on aircraft wings can decrease lift and increase the risk of crashes [2]; icing on trains can cause derailments [3]; and icing on cars can extend braking distances and compromise driving safety [4]. Therefore, the development of effective anti-icing technologies is crucial for ensuring transportation safety. To address the issue of icing on structural surfaces, researchers, including Rekuviene [5] and Zhou [6] et al., have developed both active and passive anti-icing strategies. Active anti-icing methods involve the removal of ice during its formation through external means such as mechanical, thermal, and chemical techniques. However, each method has inherent limitations. The mechanical method is labor-intensive and time-consuming, and it may damage the structural surface after repeated ice removal. The thermal method requires an additional energy supply system and results in high energy consumption. The chemical method reduces ice accumulation by spraying a special reagent on the surface to lower the freezing point of water, but the chemical reagent can corrode the structural surface and have adverse environmental effects. In contrast, passive methods utilize the physical and chemical properties of materials to prevent ice condensation or reduce the adhesion strength between ice and the surface. With this approach, ice adhering to the surface will naturally fall off due to gravity or wind, thereby consuming minimal energy from the vehicle [2].

A photothermal and hydrophobic surface with a nano-micro structure (PHS-NMS) is defined as an engineered surface that possesses hierarchical structures at both the microscale and nanoscale, designed to enhance specific properties such as hydrophobicity and photothermal conversion efficiency. These surfaces are fabricated to achieve a water contact angle (*θ*) of at least 90°, indicating hydrophobicity while also exhibiting definite photothermal conversion capabilities. The nano-micro structures manipulate the interaction between the surface and liquids or light, optimizing the surface properties for applications in photothermal and hydrophobic materials [7]. In recent years, as a key material in passive anti-icing strategies, PHS-NMSs have provided new insights into achieving efficient and environmentally friendly anti-icing due to their combined optical absorption, thermal conversion capabilities, and excellent hydrophobicity. PHS-NMSs can absorb visible and near-infrared light from the sun, converting the light energy into heat. This mechanism elevates the surface temperature, thereby suppressing ice nuclei formation, delaying ice layer growth, and even inducing ice layer melting under sunlight. On the other hand, the nano-micro structure of these surfaces facilitates the formation of gas–solid–liquid contact states with external droplets. The surface’s nano-micro roughness traps air, creating an air layer at the liquid–solid interface, which reduces the contact area between droplets and the solid surface. Consequently, heat transfer efficiency is diminished, delaying droplet freezing and preventing ice crystal formation on the surface [8]. Additionally, the reduced contact area between the solid surface and the liquid droplets decreases the interaction force between the rough structured surface and the droplets, allowing the droplets to roll off the surface when tilting the substrate at a certain angle, thus avoiding ice condensation [9].

In line with the global trend towards low-carbon and energy-efficient practices, a PHS-NMS consumes less energy and aligns with the concept of sustainable development. Therefore, a PHS-NMS holds substantial promise in the field of anti-icing. This review systematically summarizes recent advancements, prevalent fabrication techniques, and potential applications of PHS-NMSs. It further supports the development and application of photothermal and hydrophobic materials alongside the fabrication of nano-microstructures on diverse material surfaces.

## 2. Methods

This review aims to systematically summarize the recent advancements in photothermal and hydrophobic surfaces with nano-micro structures (PHS-NMSs) for anti-icing applications. The primary objectives are to (1) elucidate the mechanisms of ice prevention, (2) describe the fabrication methods of PHS-NMSs, (3) explore pathways for performance optimization, and (4) evaluate the anti-icing performance of PHS-NMSs in various application scenarios.

Studies were eligible for inclusion if they (1) focused on photothermal and hydrophobic surfaces with nano-micro structures for anti-icing applications; (2) were published in English or with an English abstract; (3) were peer-reviewed journal articles, review articles, or book chapters; and (4) provided detailed information on material fabrication, performance evaluation, or application case studies. Exclusion criteria included (1) non-peer-reviewed articles (e.g., conference proceedings, preprints) and (2) studies unrelated to anti-icing applications.

A comprehensive literature search was conducted using the following databases: Web of Science, Scopus, and Google Scholar. The search terms included “photothermal surfaces”, “hydrophobic surfaces”, “nano-micro structures”, “anti-icing”, and “fabrication methods”. The search was limited to references published between January 2010 and December 2024. The search was conducted in January 2025.

The selection process involved two stages: (1) initial screening based on titles and abstracts to identify potentially relevant studies and (2) full-text evaluation to assess eligibility according to the inclusion and exclusion criteria. Two reviewers independently conducted the screening and selection process. Discrepancies were resolved through discussion and consensus.

Data extracted from eligible studies included (1) study objectives and background, (2) types of materials and nano-micro structures used, (3) fabrication methods, (4) performance evaluation metrics (e.g., photothermal conversion efficiency, hydrophobicity, and anti-icing performance), and (5) application scenarios. The extracted data were organized into tables and discussed in the context of the review objectives.

The quality of the included studies was assessed based on the clarity of research objectives, appropriateness of methods, validity of results, and relevance to anti-icing applications. Studies with detailed experimental procedures, clear performance metrics, and relevant application case studies were considered high quality.

## 3. Photothermal Materials and Their Photothermal Conversion Properties

### 3.1. An Overview of the Types of Photothermal Materials and Their Photothermal Conversion Mechanisms

Currently, commonly used photothermal materials include carbon photothermal materials, polymer photothermal materials, semiconductor photothermal materials, and metal photothermal materials. Their photothermal conversion mechanisms are as follows:(1)Carbon photothermal materials. The photothermal conversion of carbon photothermal materials is attributed to the material’s ability to absorb light energy and convert it into heat through lattice vibrations. When the incident light energy matches the energy required for electron transitions, the electrons absorb the incoming light and jump from the ground state to higher energy orbitals. The excited electrons relax through electron–phonon coupling, and the absorbed light energy is transferred from the excited electrons throughout the atom via lattice vibrations, leading to an increase in the material’s temperature [10]. Common carbon photothermal materials include carbon nanotubes, graphite, graphene, graphene oxide/reduced graphene oxide, and carbon black [11,12].(2)Polymer photothermal materials. When polymer photothermal materials are irradiated by incident light, and the photon energy matches the possible electron transitions within the molecule, the excited electrons jump from the lowest unoccupied molecular orbital to the highest occupied molecular orbital. Energy is released when the excited state electrons relax back to their ground state, leading to an increase in the material’s temperature [13]. Common polymer photothermal materials include polypyrrole, polyaniline, and polydopamine.(3)Semiconductor photothermal materials. The absorption of light by semiconductor photothermal materials primarily depends on intrinsic absorption (band gap absorption). When incident light strikes these materials, electron–hole pairs are generated. The excited electrons eventually return to a lower energy state, releasing energy either as radiative relaxation in the form of photons or non-radiative relaxation in the form of phonons. When the energy is released as phonons, it leads to localized warming of the lattice, establishing a temperature distribution that depends on light absorption and recombination properties, thereby giving rise to the photothermal effect [14,15]. Common semiconducting photothermal materials include hydrogenated black titanium dioxide (Black TiO_2_), Ti_2_O_3_ nanoparticles, and Fe_3_O_4_.(4)Metal photothermal materials. Metals can be prepared as photothermal materials due to the localized surface plasmon resonance (LSPR) effect produced by metal particles on their surfaces [16]. When the vibration frequency of the incident photons matches the vibration frequency of the electrons on the metal surface, plasmon resonance occurs, causing collective excitation of the electrons. The excited hot electrons resonate with the incident electromagnetic field, causing the free electrons inside the metal particles to convert kinetic energy into heat and radiate electromagnetic waves through damping, resulting in a rapid increase in the material’s surface temperature [17]. Common metallic photothermal materials include Au, Ag, Al, Ni, and Cu.

Understanding the fundamental photothermal conversion mechanisms of these materials is crucial for optimizing their performance in practical applications. In the context of photothermal and hydrophobic surfaces with nano-micro structures (PHS-NMSs), the selection of appropriate photothermal materials plays a pivotal role in enhancing solar light absorption and photothermal conversion efficiency, which are essential for effective anti-icing applications. This leads us to a detailed discussion on the photothermal properties of surfaces with nano-micro structures.

### 3.2. Photothermal Properties of Surfaces with Nano-Micro Structures

In the development of PHS-NMSs, the selection of photothermal materials primarily depends on their solar light absorption capacity and photothermal conversion performance. The solar light absorption capacity is mainly assessed by the material’s solar spectral absorption range and the light absorption intensity at each wavelength. Figure 1 presents the solar spectral map, where solar radiation is predominantly concentrated in the 200~2500 nm wavelength range. Approximately 3% of the solar radiation energy is in the ultraviolet wavelength band (200~400 nm), about 45% is in the visible wavelength band (400~760 nm), and roughly 52% is in the near-infrared wavelength band (760~2500 nm) [18]. Solar absorptivity is an indicator used to evaluate a material’s ability to absorb solar light and is defined as the ratio of the solar radiation absorbed by the material to the total incident radiation. The higher the solar absorptivity of a material, the greater its light absorption capability. To enhance solar absorptive capacity, a photothermal material should exhibit maximum absorptivity and minimum reflectance and transmittance in the energy-rich spectral range of solar radiation (400~760 nm).

In addition to light absorption, photothermal conversion performance is a critical factor in evaluating photothermal materials. Materials with excellent photothermal performance can efficiently convert absorbed sunlight into heat rather than other forms of energy [18]. To quantify the photothermal conversion performance of a material, the photothermal conversion efficiency can be calculated by measuring the temperature rise on the material’s surface and the energy output of the incident light source [19]. In the application of photothermal materials, researchers often optimize the material’s surface structure to construct light traps on a nanometer to micron scale, thereby enhancing light absorption and photothermal conversion performance. Common light trap structures include groove structures, porous structures, array structures, and layered structures [11]. Notably, Ren et al. [20] fabricated graphene foams with a porous layered structure via plasma-enhanced chemical vapor deposition. These foams exhibit the ability to absorb the full solar spectrum and achieve photothermal conversion efficiency with a light absorption rate of 93.4%. Zhou et al. [21] employed a physical vapor deposition system to deposit gold nanoparticles on anodic aluminum oxide surfaces with varying pore sizes. This approach yielded photothermal materials with superior conversion performance, characterized by a light absorption rate exceeding 90%. It has been demonstrated that nano-micro structures play a crucial role in improving photothermal conversion performance by promoting the absorption, conversion, and utilization of light energy through mechanisms such as light trapping and local enhancement effects, heat transfer and distribution optimization, and performance synergy. This enhancement in photothermal conversion efficiency shows broad application potential in various fields [22].

Despite the significant photothermal conversion efficiencies achieved in the abovementioned studies, a comprehensive comparison of photothermal conversion efficiency, power density, and surface temperature rise for photothermal surfaces with nano-micro structures remains challenging due to limitations in existing research data. Table 1 lists the photothermal conversion efficiencies of some photothermal nano-micro surfaces after power density normalization.

To date, PHS-NMSs have been significantly driven by the meticulous selection and optimization of photothermal materials, as well as the ingenious design of nano-micro structures that augment solar absorption and hydrophobicity. These surfaces have emerged as promising candidates for anti-icing applications, capitalizing on their distinctive optical and thermal attributes to effectively inhibit ice formation. Nevertheless, the successful and scalable implementation of PHS-NMSs hinges not only on their intrinsic material properties but also on the robustness and reproducibility of the fabrication processes that create these surfaces.

With this in mind, the subsequent section will delve into the diverse array of methods employed to fabricate hydrophobic surfaces with nano-micro structures, elucidating their specific anti-icing properties. This exploration aims to shed light on the practical considerations involved in the synthesis of these sophisticated surfaces while also spotlighting the challenges and opportunities that arise when scaling up these technologies for widespread application.

## 4. Preparation of a Hydrophobic Surface with a Nano-Micro Structure and Its Anti-Icing Properties

### 4.1. Wetting Theory of Hydrophobic Surfaces with Nano-Micro Structures

The lotus leaf exhibits a self-cleaning function, which is attributed to its hydrophobicity. It is caused by a wax composition with a low free energy and unique nano-micro structure. Since Barthlott et al. [25] first observed the surface of the lotus leaf using a scanning electron microscope in 1997 and revealed the relationship between the surface’s nano-micro structure and the lotus leaf’s “self-cleaning” effect, a global research boom on hydrophobic surfaces has been initiated.

When a water droplet contacts a surface, it forms a droplet with a specific shape due to the interfacial tensions at the gas–solid–liquid three-phase boundary, as well as gravity. The contact angle, *θ* is defined as the angle between the tangent to the liquid surface at the point of contact and the tangent to the solid surface at the same point. If the contact angle *θ* of water droplets on a material’s surface exceeds 90°, the surface is considered hydrophobic; if *θ* is greater than 150°, it is classified as superhydrophobic [26]. The hydrophobicity of surfaces with certain nano-micro structures can be theoretically well explained [27]. Based on Young’s equation [28], which describes the relationship between the water contact angle and surface energy at the solid–liquid–gas three-phase interface, the Wenzel model [29] and the Cassie–Baxter model [30] were subsequently developed. These models are widely acknowledged for their ability to explain the hydrophobic behavior observed on surfaces with intricate structures [31], as depicted in Figure 2. The actual wettability of surfaces typically lies between the extremes described by the Wenzel and Cassie–Baxter models, a state referred to as the mixed Cassie–Wenzel model (Figure 2), where the liquid partially wets the rough surface structure [32]. In such cases, the contact angle can be expressed by Equation (1).(1)$$\cos\theta_{C-W}=f_{1}\left(r\cos\theta_{\gamma }+1\right)-1$$
where *θ_C−W_* represents the apparent contact angle, *f*_1_ is the proportion of the contact area between the liquid droplet and the rough surface, and *r* denotes the surface roughness. Building upon the foundation of low surface energy, this model further introduces a key parameter of hydrophobicity, the small solid–liquid contact area, which can be achieved by constructing the nano-micro structure on the surface.

The rolling angle, *ϕ* is commonly used to quantify the dynamic behavior of a water droplet on a solid surface and is defined as the angle of surface inclination at which the droplet begins to roll. A smaller rolling angle indicates weaker adhesion of the water droplet to the surface. If a material surface simultaneously satisfies the conditions of *θ* > 150° and *ϕ* < 10°, it will exhibit a unique self-cleaning effect similar to that of a lotus leaf. In this superhydrophobic low-adhesion state, water droplets encounter minimal resistance on the surface and can pick up surface contaminants as they roll away. It has been demonstrated that hydrophobic surfaces possess some anti-icing capabilities, which delay the onset of icing, reduce the area of ice formation, and decrease the adhesion force of ice to the solid surface [25,33,34,35]. This anti-icing property of hydrophobic surfaces makes them valuable for various applications, such as aircraft, wind turbines, outdoor antennas, and power grids.

### 4.2. Preparation of Hydrophobic Surfaces with Nano-Micro Structures

Previous studies have demonstrated that the hydrophobic state of a surface is achieved through a combination of two key factors: low surface free energy and unique surface structure. While achieving low surface free energy is relatively straightforward, the development of nano-micro structures that can produce hydrophobic or even superhydrophobic surfaces, as well as the stability and performance of these materials for subsequent applications, has become the primary research focus in this field. To date, researchers have developed a variety of processes to create surfaces with nano-micro structures.

(1)Surface Modification Techniques
(1)Physical removal methods. These include plasma treatment [36], electron beam etching [37], photolithography [38], and other techniques. The common feature of these methods is the construction of nano- and/or microscale rough topography by physically removing material from the surface. These approaches are typically costly and inefficient and are more prevalent in basic research.(2)Chemical growth and etching. These encompass thermal oxidation [39], chemical vapor deposition [16], electrochemical deposition [40], chemical etching [41], and others. The common feature of these methods is the formation of nano-micro structures through various chemical reactions on the surface, resulting in deposition, growth, or corrosion. These methods are some of the primary means of preparing PHS-NMSs currently. However, they have drawbacks such as long reaction times and poor surface structure homogeneity, which hinder the production of large-area samples with high consistency. Additionally, the use of various chemicals can lead to environmental pollution. Moreover, the adhesion of the surface nano-micro structures to the substrate is often weak, making them susceptible to damage.(3)Coatings. The self-stacking of nanoparticles on a surface is an effective means of achieving nano-micro structures, and recent years have seen increased research in this area. Different nanoparticles possess varying properties, allowing for the construction of various specialized wettable surfaces. Various types of low free energy-modified nanoparticles can be used to prepare hydrophobic and superhydrophobic coatings [42]. The core issues in coating methods include the effective dispersion of nanoparticles, as well as the adhesion and film-forming properties of the coatings.(4)Electrostatic spinning. During the electrostatic spinning process, a polymer solution or melt forms a jet under the influence of a strong electric field. As the solvent evaporates or the melt cools, the jet solidifies into fibers, which are then deposited on a collector to form a two-dimensional membrane material with a fiber structure ranging from micrometers to nanometers. These fibers interweave to create a porous network structure characterized by high porosity and specific surface area, effectively trapping air and thus exhibiting hydrophobic properties [43]. By selecting different raw materials and adjusting the parameters of the electrostatic spinning process, the diameter, orientation, and pore structure of the fibers can be controlled, thereby optimizing their hydrophobic properties. The electrostatic spinning method is relatively simple to operate, cost-effective, and scalable for large-area production, but the homogeneity and stability of the resulting fiber structure need further improvement, and the mechanical properties of the fiber membrane are relatively weak.(2)Replication and Fabrication Techniques
(1)Replica imprinting. Obtaining micro- and nanostructures similar to those found on plant surfaces through organic polymer replication is a widely used approach. A single replica can produce a negative structure corresponding to the surface structure, while two replicas can yield a micro- and nanostructure that closely approximates the original surface [44,45]. However, these methods are generally limited to polymeric materials and have limited capability for replicating complex nanostructures.(2)Femtosecond laser processing. Femtosecond laser processing is an emerging technique for fabricating nano-micro structures. It utilizes ultrashort pulsed laser light to interact with materials, inducing the formation of nano-micro structures on the surface by controlling parameters such as laser energy, pulse width, and scanning speed. This method offers advantages such as high precision, high resolution, and a small heat-affected zone, enabling the fabrication of regular and fine structures such as column arrays and grooves on a variety of materials, including metals, semiconductors, ceramics, and polymers [46,47]. For instance, on metal surfaces, femtosecond lasers can induce periodic micro- and nanoscale raised structures, thereby altering surface wettability and achieving hydrophobicity. However, the high cost of femtosecond laser processing equipment, low processing efficiency, and the expense associated with large-area processing limit its widespread application.(3)3D printing. Three-dimensional surfaces with specific microstructures can be constructed by precisely controlling the layer-by-layer deposition of 3D printing materials. In the preparation of hydrophobic surfaces, designs can incorporate layered structures, porous structures, or rough textures that trap air and form hydrophobic interfaces. For example, 3D printing with low surface energy polymer materials allows for the direct fabrication of complex-shaped workpieces with superhydrophobic properties [48]. However, 3D printing technology currently has limited resolution for microstructure fabrication, and there are significant challenges in achieving controlled preparation of sub-micron and nanoscale structures due to slow printing speeds and relatively limited material options.

The aforementioned methods each have distinct advantages and disadvantages, making them suitable for different substrate materials and application areas, with the resulting surface micro- and nanostructures exhibiting unique characteristics. Physical removal methods, such as plasma treatment, electron beam etching, and photolithography, are effective for creating micrometer or submicrometer structures with precise control over structure size. However, these methods generally lack the ability to regulate nanostructures. Conversely, chemical growth, etching, and coating methods, including thermal oxidation, chemical vapor deposition, electrochemical deposition, and nanoparticle self-assembly, are more conducive to producing nanoscale rough structures such as nanowires, nanosheets, and nanoparticles. These methods, however, do not provide the ability to regulate micron-scale structures effectively [49]. Femtosecond laser processing offers exceptional precision for the fabrication of both nano- and microstructures, enabling the creation of highly regular and fine surface architectures. However, its practicality is constrained by high operational costs and limited scalability for large-area applications [50]. Three-dimensional (3D) printing demonstrates the capability to construct intricate three-dimensional architectures with specific microstructures, such as layered or porous designs. However, its resolution remains a critical limitation, especially for sub-micron and nanoscale structures [51]. Electrospinning enables the scalable production of nanofiber structures over large areas, with the ability to create porous networks that exhibit hydrophobic properties. However, challenges related to the uniformity and structural stability of the resulting fibers persist and require further optimization [52]. In practical applications, it is often necessary to select the appropriate preparation method or combine multiple methods to create ideal hydrophobic surfaces with nano-micro structures based on specific needs.

### 4.3. Anti-Icing Properties of Hydrophobic Surfaces with Nano-Micro Structures

In this study, we adopted a comprehensive and integrated approach to analyze hydrophobic and superhydrophobic surfaces within a unified framework. This method is based on the fundamental principles of surface wettability; although their water-repellency levels differ, they all stem from the basic mechanisms of surface chemistry and micro/nanoscale topography. Therefore, the hydrophobic surfaces with nano-micro structures mentioned below actually include cases where the surface exhibits a superhydrophobic state. Current research has provided preliminary evidence that hydrophobic surfaces with nano-micro structures play three potential roles in the anti-icing process:(1)Avoiding adherent aggregation of supercooled water. When supercooled water droplets contact a hydrophobic surface, they spontaneously roll off because they cannot adhere, thus preventing surface icing. Cao et al. [53] prepared hydrophobic surfaces by compositing nanoparticles with polymers and found that these surfaces significantly reduced icing by preventing the accumulation of supercooled water. Under outdoor freezing rain conditions, the amount of icing on the hydrophobic surface was drastically reduced compared to that on a normal surface. Wang et al. [54] developed an organic–inorganic composite superhydrophobic surface with a typical micro- and nano-composite structure, which prevented water attachment even at −20 °C. The study by Lv et al. [55] demonstrated that above −25 °C, a water droplet impacting the prepared superhydrophobic surface bounces up spontaneously, thereby avoiding the formation of ice.(2)Delaying the nucleation and icing process of surface water droplets. This effect is currently a subject of debate [36,56]. Although certain studies have suggested that hydrophobic surfaces can significantly delay the icing process, the experimental findings of Kulinich et al. [37] reveal a more nuanced perspective. Among the 14 solid surfaces investigated, pre-treated smooth silicon surfaces demonstrated the most effective performance in delaying icing. In contrast, rough hydrophobic surfaces were found to be ineffective in impeding the icing process. In the positive findings reported, there is significant variation in the delayed icing times on hydrophobic surfaces under the same temperature conditions, ranging from 100 s to 100 min [57,58]. This potential delayed icing effect is related to two factors. First, when water droplets contact the surface, they are generally warmer than the surface, and the hydrophobic surface reduces heat exchange between the droplets and the surface, thereby delaying droplet cooling [59]. Second, most studies suggest that the delay in icing time is associated with an increased heterogeneous nucleation potential barrier at the surface [9]. However, the specific delayed icing capacity of hydrophobic surfaces cannot yet be clearly defined due to the large discrepancies in reported delayed icing times across different literature.(3)Reducing the adhesion between ice and surface. The previous roles focus on avoiding or reducing the formation of ice. However, other researchers have investigated the ability of hydrophobic surfaces to reduce ice adhesion. Studies have shown that certain hydrophobic surfaces can significantly decrease ice adhesion [60,61]. Nevertheless, this effect tends to diminish with an increasing number of icing–melting cycles [62,63]. Additionally, influenced by environmental factors, ice adhesion on hydrophobic surfaces can increase under certain conditions. For instance, Chen et al. [64] found that ice adhesion on hydrophobic surfaces can be approximately five times higher than that on ordinary surfaces. Currently, some researchers argue that a hydrophobic surface does not necessarily imply an ice-repellent surface, and the two are not necessarily related [65]. In the study of hydrophobic anti-icing properties, the influence of surface frosting has begun to attract attention. In low-temperature environments, supersaturated water vapor in the air continuously condenses and forms frost on cold surfaces. This frosting process may affect the adhesion and ice formation of external liquids, as well as significantly impact ice adhesion [34,66]. At present, most relevant studies only mention this phenomenon, and there is a lack of systematic research on its effects on the icing process.

The hydrophobic anti-icing effects of surfaces in the above studies were generally obtained in the laboratory or under static freezing conditions, and a few studies on dynamic anti-icing of surfaces have also been conducted. In recent years, there have been many reports of related studies based on ice wind tunnels.

In 2011, Antonini et al. [67] elucidated the active anti-icing mechanism of a synergistic anti-icing system incorporating nano-micro-structured surfaces through experimental studies conducted in an ice wind tunnel. Their findings indicate that water droplets on hydrophilic surfaces exhibit slow movement, leading to icing if the droplets fail to detach from the surface under the influence of external forces. Conversely, on hydrophobic surfaces, water droplets demonstrate rapid rebound, splashing, and detachment from the surface when subjected to external forces, thereby effectively mitigating icing. In the same year, the experimental study of Fortin et al. [68] showed that materials with nano-micro-structured surfaces can save 33% and 13% of the energy consumption of the anti-icing system under light ice and frost ice conditions, respectively, whereas ordinary surfaces brushed with hydrophobic coatings can save 13% and 8% of the energy consumption of the anti-icing system under light ice and frost ice conditions, respectively. In 2016, Waldman et al. [69] showed that airfoils with different types of nano-micro-structured surfaces exhibit different icing processes and form significantly different ice patterns under the same icing conditions. Mishchenko et al. [56] used a high-speed video camera to observe the process of supercooled water at −10 °C impacting different surfaces and found that water droplets would rapidly spread and freeze on a normal surface. On the other hand, on hydrophobic surfaces, the water droplets would continuously bounce off and eventually leave the surface. The study by Boinovich et al. [60] found that at a low temperature of −15 °C, the water droplets on ordinary surfaces would nucleate and freeze within 40 min, while for hydrophobic surfaces, this time can be delayed for more than 2 h. The study also showed that the water droplets on the surface can be frozen in a short time. Table 2 lists the anti-icing properties of hydrophobic surfaces with nano-micro structures in ice wind tunnels.

All of the above studies used a coating-type approach in the preparation of nano-micro-structured hydrophobic surfaces. Although the coatings varied, they were all realized by adding nano-micro particles to the low free energy coatings. All these studies show that icing mainly originates from the heterogeneous nucleation process of supercooled water on solid surfaces, and this effect of avoiding supercooled water collection at low temperatures, which is possessed by hydrophobic surfaces, can greatly reduce or even inhibit icing.

Having examined the diverse fabrication methods and anti-icing properties of hydrophobic surfaces with nano-micro structures, we have highlighted their significant potential for practical applications. These methods, ranging from chemical synthesis to advanced nanostructuring techniques, have demonstrated remarkable versatility in producing surfaces with tailored hydrophobic and anti-icing characteristics. However, the true potential of these surfaces can only be fully realized by understanding their performance in specific applications and addressing the challenges associated with material stability, scalability, and environmental adaptability. Building on this foundation, the subsequent discussion will delve into the preparation and anti-icing properties of PHS-NMSs across different material categories, including carbon-based, polymer-based, semiconductor-based, and metal-based composites. This analysis will also provide insights into the unique advantages and limitations of each material type, as well as their potential applications in various industrial and environmental contexts. By examining these specific types of PHS-NMSs, we aim to offer a comprehensive overview of the current state of research and identify promising directions for future development.

## 5. Preparation and Anti-Icing Properties of PHS-NMSs

### 5.1. Carbon-Based PHS-NMSs

Carbon-based photothermal hydrophobic surfaces with nano-micro structures (PHS-NMSs) are materials that exhibit both photothermal conversion and hydrophobic properties constructed on a carbon base. Common raw materials for this type include carbon nanotubes, graphite, graphene, graphene oxide/reduced graphene oxide, and carbon black, which are used to achieve photothermal hydrophobic functionality through compounding with other substances or through special preparation processes. Wu et al. [70] utilized poly(dimethylsiloxane) (PDMS), SiO_2_ shells, and candle ash (CS) to create a low-cost, high-efficiency composite surface, referred to as the PSCS surface. This surface can reach a temperature of 53 °C at 1× solar intensity, and droplets do not freeze on the surface at ambient temperatures as low as −50 °C. Additionally, it can rapidly melt frost and ice buildup on the surface within 300 s. Li et al. [71] prepared a photothermal superhydrophobic coating using carbon black nanoparticles as the photothermal material by spraying. This coating could reach a surface temperature of 75.3 °C under 1× sunlight irradiation, and a 3 mm ice layer on its surface could melt and keep the surface dry within 540 s. Xie et al. [72] prepared photothermal hydrophobic materials with a layered structure using a chemical deposition method based on carbon cloth as a substrate. These materials exhibited a light absorption rate of up to 99%, and their feasibility was verified through anti-icing and frost layer melting experiments conducted on wind turbine blades (Figure 3).

Wu et al. [73] synthesized poly(dimethylsiloxane)/reduced graphene oxide (PDMS/RGO) films using a dual-template method. These films demonstrated a long freezing delay time (t_D_ > 2 h) and ultra-high de-icing efficiency (>1.05 kg m^−2^ h^−1^) under 1× solar illumination. Guo et al. [74] developed a coating with good anti-icing and photoresponsive properties. Zhang et al. [75] developed superhydrophobic carbon nanotubes/silicon dioxide/epoxy resin (CNTs-SiO_2_/Epoxy resin) coatings with excellent photothermal conversion capabilities. The surface droplet freezing time was extended by 11 times compared to ordinary epoxy resin coatings, and the ice layer on the surface (at −20 °C) could be completely removed after 60 s of near-infrared irradiation (1 W cm^−2^), whereas the ice layer on the surface of ordinary epoxy resin coatings failed to melt after 300 s under the same conditions. By integrating hydrophilic polyvinylpyrrolidone (PVP) and low-surface-energy polydimethylsiloxane (PDMS) with photothermal carbon nanofibers (CNFs), which exhibit rapid photothermal conversion capabilities, a composite material with enhanced performance can be developed. Yang et al. [76] demonstrated that a photothermal hydrophobic surface (CPS) based on candle soot could maintain a temperature of 95 °C after 30 s of illumination at 1 kW m^−2^. Due to its superhydrophobicity, the CPS exhibited superior defrosting and ice-melting effects compared to a normal matte black paint surface (BS) at −30 °C (Figure 4).

Yu et al. [77] used a simple template method to construct a porous hydrophobic dry gel and obtained a photothermal superhydrophobic surface (PMX) by incorporating multi-walled carbon nanotubes. The porous channels in the material reduce heat loss, enabling rapid warming and efficient anti-icing in low-temperature environments. Droplets on the PMX do not freeze at −30 °C under weak illumination (0.25 times solar illumination), and the surface temperature can reach 39 °C under 1× solar irradiation. When investigating the outdoor performance of the material (i.e., ambient temperature below −12 °C), it was found that accumulated snow on the PMX could be completely melted after 8 min under 0.55× solar irradiation. Xu et al. [78] successfully prepared thin films with condensing microdroplet self-propulsion (CMDSP) by constructing zinc oxide (ZnO) nano-needle structures on the surface of a carbon nanotube thin film using electrochemical methods. The prepared films exhibit strong light absorption and high energy transfer efficiency, resulting in excellent photothermal anti-icing properties. Under light irradiation of 4406 Lux, the icing delay time is over 4 h, even at −5 °C. Jiang et al. [7] prepared photothermal hydrophobic coatings with a multilayer structure by combining polyvinylidene fluoride and CNTs using a spraying method. The surface temperature of the coatings increased by more than 40% at ~1 times the solar light intensity at −20 °C. The heat transfer network formed by the coatings can rapidly melt the ice layer covering the surface (see Figure 5).

Xie et al. [79] used a template-spraying method to prepare a photothermal anti-icing material containing a carbon-based light-absorbing layer (P@MNS) with an average light absorption rate of up to ~98%. This allowed the covered frost/ice layer to melt during photothermal ice melting at 100 mW/cm^2^ light for 300 s after melting and detaching from the wall. Additionally, the material’s surface had a low freezing temperature (−25.20 ± 1.34 °C) and a long icing delay time (774.76 ± 114.19 s). Studies targeting the applied properties of such materials have also shown that the stripping of self-polymerized reinforced carbon nanomaterials can be achieved using dopamine hydrochloride, resulting in superhydrophobic surfaces with good photothermal properties and self-healing effects [80].

These studies represent the state-of-the-art in carbon-based PHS-NMSs, achieving remarkable photothermal efficiency and hydrophobicity. They effectively address critical challenges such as maintaining performance at low temperatures and enabling rapid ice removal. However, existing fabrication methods often involve complex processes and rely on expensive materials, which hinder scalability and widespread application. Future research should focus on simplifying fabrication techniques and enhancing material durability to improve practical applicability and commercial viability.

### 5.2. Polymer-Based PHS-NMSs

Polymer-based PHS-NMSs refer to materials that possess photothermal conversion and hydrophobic properties with nano-micro surface structures, using polymers as the main body and incorporating components with photothermal effects or through chemical modification. Common raw materials for this type include polypyrrole, polyaniline, and polydopamine, among others. Liang et al. [81] prepared hydrophobic surfaces with photothermal effects using polypyrrole (PPY) and fluorosilicone resin with silica (SiO_2_) particles. Due to the photothermal effect of PPY, the surface temperature of the material can reach 80 °C under 1× solar light intensity irradiation. It also exhibits excellent anti-icing performance at low temperatures of −40 °C, with a delayed icing time more than three times that of untreated samples. Xie et al. [82] prepared non-fluorinated photothermal hydrophobic coatings by oxidative polymerization of pyrrole (PPY/ATP@hexadecyIPOS, see Figure 6). The layered nano-microstructure increases the surface area of polypyrrole under sunlight, demonstrating good anti-icing performance in both laboratory and outdoor environments.

The studies showcase the potential of polymer-based materials to achieve high photothermal efficiency and hydrophobicity through simple and scalable fabrication methods. However, these materials often suffer from relatively low photothermal conversion efficiency and limited thermal stability compared to other material systems. Future developments should focus on improving the photothermal performance and durability of polymer-based PHS-NMSs while maintaining their ease of processing and tunability. This will require innovations in material design and fabrication techniques to enhance their practical applicability and commercial potential.

### 5.3. Semiconductor-Based PHS-NMSs

Semiconductor-based PHS-NMSs are materials that use a semiconductor material as the core, combined with hydrophobic components or surface modifications to achieve nano-micro surface structures with photothermal conversion and hydrophobic functions. Common semiconductor core materials include hydrogenated black titanium dioxide, Ti_2_O_3_ nanoparticles, Fe_3_O_4_, and Co_3_O_4_, among others. Yin et al. [83] incorporated Fe_3_O_4_ particles into hydrophobic coatings, resulting in the rapid melting of ice layers on the material’s surface under near-infrared light irradiation. Wu et al. [84] blended fluorinated epoxy resin with Fe_3_O_4_ nanoparticles using spray coating. The resulting hydrophobic coating with photothermal effects, obtained by the spraying method, exhibited a temperature increase of 10 °C after 1 min of 80 W infrared light irradiation, and the ice layer on the material’s surface was completely melted after 8 min of light irradiation.

Hu et al. [85] prepared hydrophobic SiO_2_/SiC coatings on an iodine-doped substrate, which can rapidly increase the surface temperature to about 200 °C, extending the freezing time of droplets to 326 s under light irradiation. Xie et al. [86] prepared photothermal hydrophobic coatings on a copper mesh using SiC particles as a photothermal conversion material and hydrophobic SiO_2_ particles as a low-surface-energy material by the spraying method. Under near-infrared irradiation, the surface temperature of the copper mesh with an ice layer can be increased to 35.3 °C in 220 s, allowing the ice layer on the surface to melt quickly. Inspired by wheat leaves, Zhang et al. [87] used ultrafast pulsed laser deposition to prepare photothermal hydrophobic materials with layered nano-micro structures. These materials exhibit excellent photothermal capabilities due to the formation of photothermal traps by iron oxide nano-micro structures, and they can efficiently remove condensed water through polymerization-induced droplet hopping, thus keeping the surface dry (with 72% of the area remaining dry), and the surface temperature can reach 5.5 °C after 1400 s at an ambient temperature of −50 °C. Liu et al. [88] created a hierarchal surface structure with waxberry-like micro-nano-structured Co_3_O_4_, which conforms to the Cassie–Baxter model theory for superhydrophobic property and 92.7% light absorptivity. Excellent anti-icing performance under a high-humidity environment was obtained, where ice delay time can be enhanced to ≥1100 s and the melting time is only 156 s under 1 sun irradiation (see Figure 7).

These studies exemplify the high photothermal efficiency and tunable hydrophobicity achievable with semiconductor materials, especially through a nano-micro structural design. However, the performance of these materials is highly dependent on interband absorption processes and thermal relaxation kinetics, which can be challenging to optimize. Future developments should focus on enhancing the stability and scalability of semiconductor-based PHS-NMSs. This includes exploring new materials with broader absorption spectra and improving fabrication methods to achieve uniform nano-micro structures, thereby maximizing photothermal conversion efficiency and hydrophobicity for practical anti-icing applications.

### 5.4. Metal-Based PHS-NMSs

Metal-based PHS-NMSs are materials that possess nano-micro surface structures and photothermal hydrophobic properties. These materials are typically prepared using metals or metal compounds as the primary photothermal conversion components, combined with hydrophobic surface modification or structural design. Metals with photothermal properties or metal compounds, such as titanium nitride (TiN) and CuFeMnO_4_, are generally used to construct photothermal hydrophobic coatings or films. Ma et al. [89] prepared selectively absorbing photothermal surfaces by chemically etching aluminum surfaces and then spin-coating TiN photothermal nanoparticles, achieving droplet nonfreezing at very low temperatures of −60 °C (Figure 8). Dash et al. [90] prepared microcosmears by utilizing metal compounds as light absorbers to create nano-micro-structured photothermal hydrophobic materials. These materials can absorb 95% of incident light and lose only 3% when re-radiated. The surface has a sufficiently fast thermal response, making the heating process faster than the freezing process, thus achieving complete anti-icing.

Wang et al. [91] prepared a nano-micro-structured photothermal hydrophobic coating (STC) by combining superhydrophobic SiO_2_ nanoparticles with CuFeMnO_4_/PDMS. Due to its high photothermal conversion efficiency, the freezing temperature of droplets on the STC surface was reduced to −35 °C, the adhesion strength of ice was decreased from 78.5 kPa (without illumination) to 12.1 kPa, and ice removal was achieved within 99 s under illumination. Zhao et al. [92] investigated the photothermal de-icing performances of smooth aluminum (S), biomimetic hydrophilic aluminum (BH), and biomimetic superhydrophobic aluminum (BS). It was shown that the surface temperature of the nano-micro-structured BS (21.7 °C) was 70 times higher than that of BH (0.3 °C) under 1× light conditions, and the ice on the surface was completely melted within 4 min through the synergistic effect of the surface superhydrophobicity and photothermal properties. Li et al. [93] utilized the laser direct structuring (LDSW) technique and PFOS modification to construct a black nano-micro-structured superhydrophobic Al surface with a water droplet contact angle of 161.2° and a light absorption capacity of over 94.5% (more than 96% in the visible spectral range). Due to its excellent photothermal properties and superhydrophobicity, the surface can achieve fast frost and ice melting at −30 °C by inhibiting the nucleation and growth of ice crystals. In related research, anti-icing on curved surfaces is quite challenging (Figure 9).

The studies highlight the superior photothermal performance and rapid ice removal capabilities of metal-based PHS-NMSs. However, these materials often face challenges such as high cost, susceptibility to oxidation, and complex fabrication processes. Future developments should focus on addressing these issues by exploring cost-effective materials, optimizing LSPR effects, and simplifying fabrication methods to enhance durability and scalability. This will be crucial for advancing metal-based PHS-NMSs towards practical and widespread anti-icing applications.

### 5.5. Composite-Based PHS-NMSs

Composite PHS-NMSs tend to have complex composite processes and more compatibility issues; however, their performance designability and synergistic performance advantages are attracting increasing attention from researchers.

To synthesize the advantages of multiple photothermal materials, researchers often introduce two or more photothermal materials to construct composite nano-micro-structured photothermal hydrophobic surfaces. Jiang et al. [94] used a spraying method to coat SiC and CNTs on ethylene vinyl acetate (EVA) substrates. The larger SiC particles provided micrometer-scale peak-like structures, while the CNTs formed a nanoscale fluffy structure on the surface of the SiC particles. The resulting photothermal hydrophobic SiC/CNT coatings exhibited a micro- and nano-graded structure. This study showed that the freezing time of droplets on its surface increased by 340% compared to the uncoated surface. In de-icing experiments, the heat was quickly transferred, rapidly melting the surface ice accumulation with a photothermal conversion efficiency of 50.94%. Sun et al. [95] presented two photothermal superhydrophobic surfaces prepared on copper substrates using displacement deposition and one-step spraying methods. These surfaces, composed of metal nanoparticles and carbon nanotubes, exhibited excellent superhydrophobicity with contact angles over 150° and rolling angles below 5°. Both materials have broadband absorption, absorbing over 90% of the solar spectrum, and can rapidly respond to solar irradiation, heating the surfaces above 0 °C to delay droplet freezing and facilitate de-icing (Figure 10).

Mitridis et al. [96] prepared photothermal superhydrophobic films using Au and nano TiO_2_. The excellent photothermal conversion efficiency induced a surface temperature increase of more than 10 °C compared to the control surface, resulting in rapid ice melting within 30 s. Jin et al. [97] prepared a large-area all-carbon-based composite photothermal surface on various substrates using a spraying method with CNTs and carbon black particles, which exhibited a light absorption of over 99.9%. Guo et al. [98] reported a highly transparent photothermal hydrophobic composite coating prepared by directly spraying Cu_7_S_4_ nanoparticles incorporated into organosilica sol on a glass surface. This multifunctional coating achieved a water droplet condensation time of 86 s at −10 °C, which was 3.42 times more delayed than that of ordinary blank glass, while the adhesion strength of ice on the coated surface was reduced to 72 kPa, about one-third of the adhesion strength of ice on the ordinary glass surface. The prepared functional coated glass has stable physical/chemical properties, and its transmittance in the visible region exceeds 75%. Composite nano-micro-structured photothermal hydrophobic materials can also be obtained by a simple scrape-coating method [99], and it has been shown that the prepared materials have excellent superhydrophobic photothermal properties and good weathering resistance, with relatively excellent adhesion and low-light de-icing performances.

These studies were highlighted for their ability to demonstrate synergistic effects achieved through composite design, thereby addressing the limitations of single-material systems. However, composite PHS-NMSs face challenges such as material compatibility and complex fabrication processes. Future developments should focus on optimizing the interactions between different materials, simplifying composite fabrication methods, and improving scalability without compromising performance. This will be essential for realizing the full potential of composite-based PHS-NMSs in practical anti-icing applications.

## 6. Summary

Research on photothermal hydrophobic surfaces with nano-micro structures (PHS-NMSs) is rapidly evolving due to their unique combination of photothermal conversion and hydrophobicity. Beyond ice prevention, these materials have demonstrated significant potential for applications in seawater desalination, crude oil cleaning, photothermal disinfection, and photothermal drive, among others. However, despite substantial progress in this field, the reviewed literature reveals several critical challenges and issues that must be addressed through future in-depth studies, as follows:(1)Durability and stability. Many PHS-NMSs exhibit performance degradation when subjected to long-term use or harsh environmental conditions. For practical applications such as aerospace, wind turbines, and outdoor infrastructure, materials must maintain stable photothermal and hydrophobic properties over extended periods. Future research should focus on developing robust surface coatings or structural designs that can resist mechanical wear, chemical corrosion, and UV degradation, thereby enhancing the longevity and reliability of PHS-NMSs in real-world scenarios.(2)Cost and benefit. The ideal solar thermal hydrophobic material should possess high solar absorption rates and significant solar-to-thermal conversion efficiency and be derived from abundant and low-cost raw materials. Additionally, the preparation methods should be simple and scalable to facilitate large-scale production. Currently, some high-performance PHS-NMSs rely on expensive raw materials or complex, multi-step fabrication processes, which hinder their widespread adoption. Future work should aim to identify cost-effective materials and streamline synthesis methods without compromising performance, thus improving the economic viability of these materials for large-scale applications.(3)Environmental adaptability. The performance of PHS-NMSs can vary significantly under different environmental conditions, including extreme temperatures, humidity levels, and UV irradiation. For instance, materials that perform well in laboratory settings may fail under outdoor conditions due to changes in ambient temperature or exposure to UV light. To ensure stable operation across diverse environments, future research should focus on enhancing the environmental adaptability of PHS-NMSs. This could involve developing materials with tunable properties or incorporating protective coatings that can mitigate the effects of environmental stressors.(4)Safety and biocompatibility. The application of PHS-NMSs in sensitive fields such as biomedicine and food processing necessitates a thorough evaluation of their safety and biocompatibility. Materials intended for these applications must be non-toxic, non-immunogenic, and environmentally benign. Future studies should include comprehensive safety assessments, particularly for materials incorporating nanoparticles or novel chemical compounds. Additionally, research should explore the potential for biodegradable or eco-friendly PHS-NMSs to minimize environmental impact.

Future research should prioritize the development of multifunctional materials with exceptional photothermal and superhydrophobic properties, robust stability, and low cost. Advanced nano-micro manufacturing technologies should be employed to control material microstructures, thereby enhancing light absorption and photothermal conversion efficiencies. Key focuses include improving material stability under low-temperature and high-humidity conditions, optimizing performance under low-intensity solar irradiation (e.g., 0.1 solar radiation intensity), and achieving dual or multiple self-healing functions. Novel preparation technologies, such as structured photothermal energy storage, near-infrared photothermal superhydrophobic materials, and femtosecond laser element doping combined with low-temperature annealing, will drive progress in this field. Additionally, efforts should be directed toward enhancing material durability, exploring cost-effective preparation methods and raw materials, and reducing production costs to accelerate the commercialization of PHS-NMSs. In summary, PHS-NMSs hold significant application potential and broad development prospects. Future work should focus on systematic research in material design, preparation processes, performance regulation, and optimization to facilitate their practical applications across various fields.

## Figures and Tables

**Figure 1 nanomaterials-15-00378-f001:**
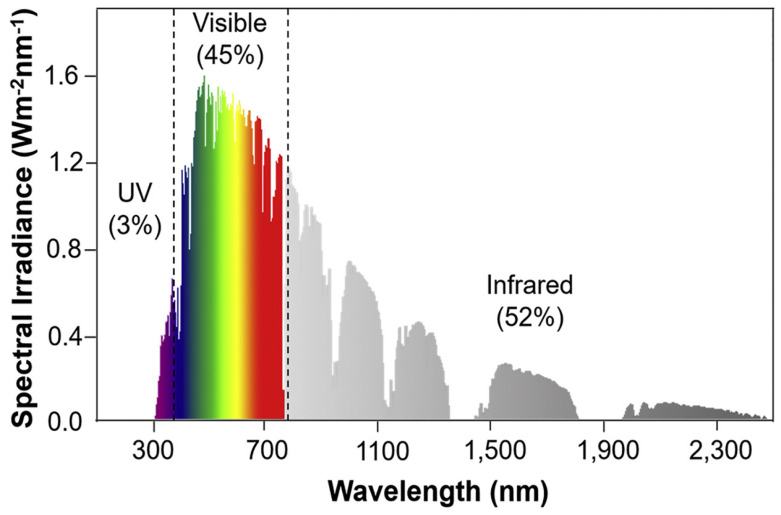
Spectrum of solar radiation on the surface of Earth (air mass 1.5 G) [18].

**Figure 2 nanomaterials-15-00378-f002:**
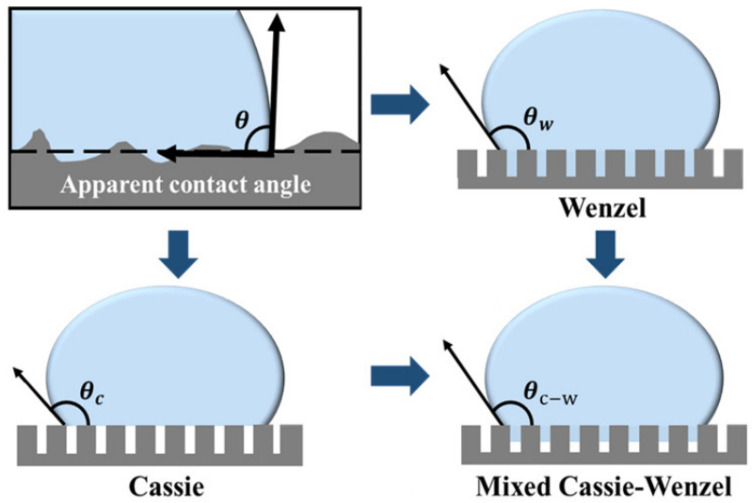
A schematic diagram of the apparent contact angle, Wenzel’s model, Cassie–Baxter’s model, and the mixed Cassie–Wenzel model [27].

**Figure 3 nanomaterials-15-00378-f003:**
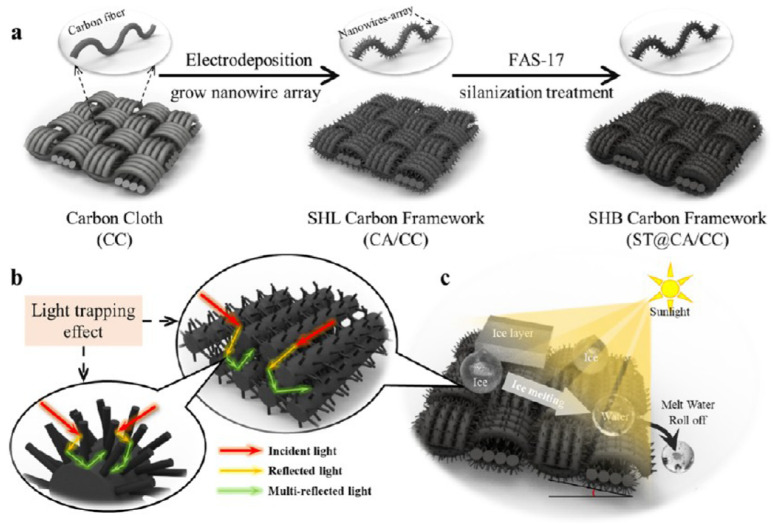
Schematic diagram of the fabrication process and photothermal anti-icing mechanism of the carbon-based photothermal superhydrophobic surfaces [72]. (**a**) Fabrication process; (**b**) Illustration of the light trapping effect induced by the nano-micro hierarchical structure; (**c**) Photothermal anti-icing mechanism.

**Figure 4 nanomaterials-15-00378-f004:**
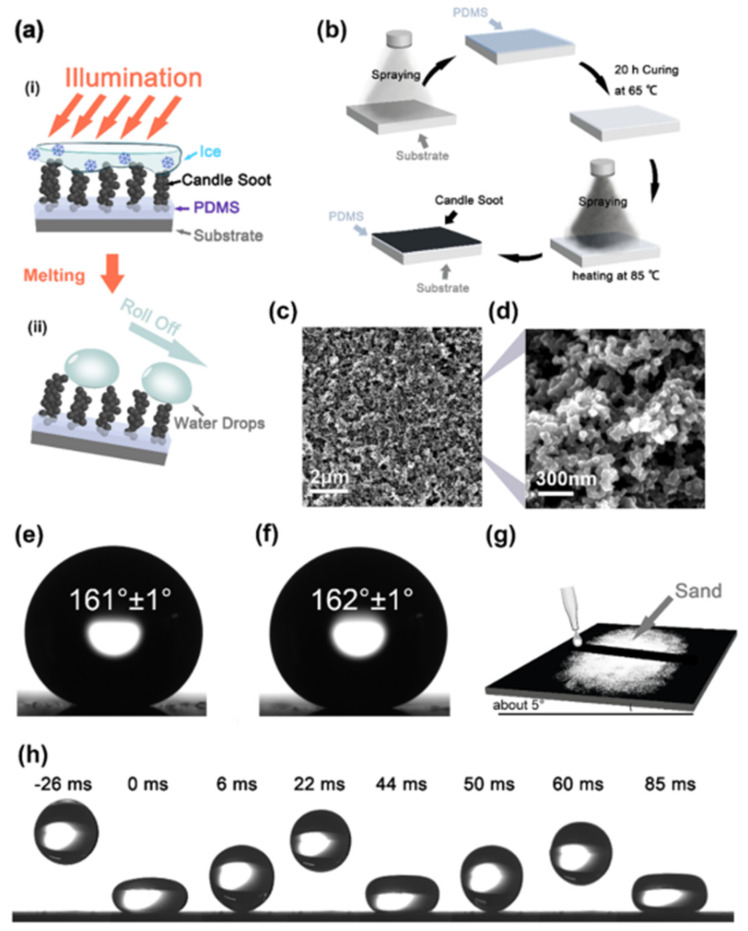
Icephobic mechanics and morphology of the CPS [76]. (**a**) The icephobic mechanism of the CPS coating: (**i**) The icing surface generates heat from the illumination, forcing the ice to melt; (**ii**) the melting water gathers into water drops, and water drops roll off. (**b**) The preparation process of the CPS. (**c**,**d**) SEM photograph of the CPS coating in different magnifications. (**e**) The water contact angle of the CNS coating; (**f**) The water contact angle of the CPS coating. (**g**) Self-cleaning experiment with the CPS. (**h**) A drop of 6 µL of water bounced on the CPS coating.

**Figure 5 nanomaterials-15-00378-f005:**
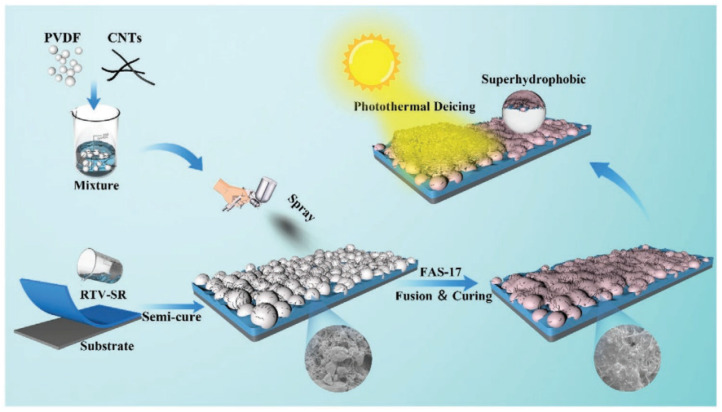
Schematic illustration of the photothermal hydrophobic coating preparation and photothermal anti-icing [7].

**Figure 6 nanomaterials-15-00378-f006:**
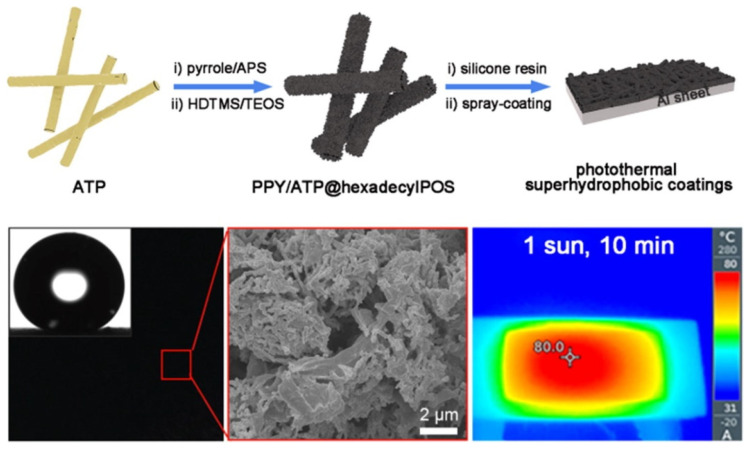
Schematic illustration of the PPY/ATP@hexadecyIPOS coating preparation process with a SEM image and the photothermal effect of the coating [82].

**Figure 7 nanomaterials-15-00378-f007:**
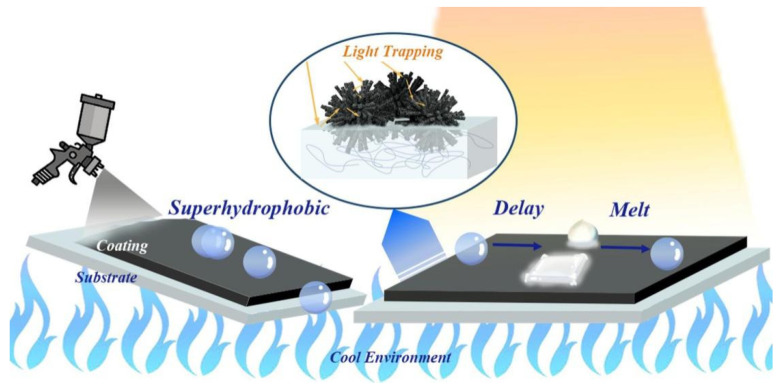
Schematic diagram of the preparation and application of a PHS-NMS with waxberry-like Co_3_O_4_ light trapping [88].

**Figure 8 nanomaterials-15-00378-f008:**
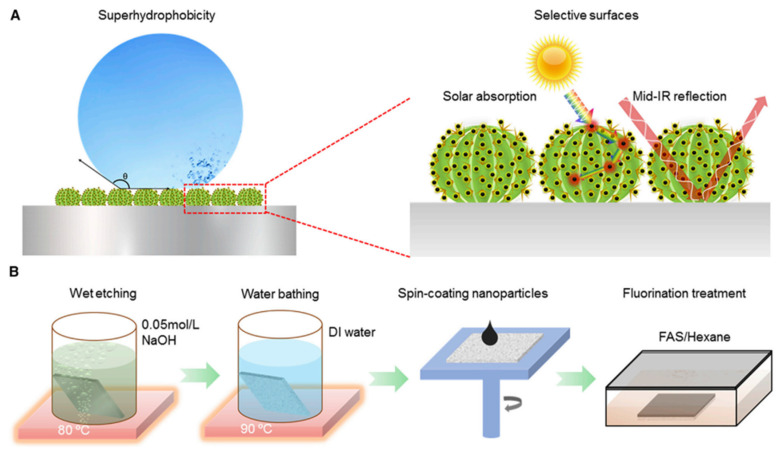
Schematic of the superhydrophobic selective solar absorber (SHSSA) design and fabrication procedures [89]. (**A**) Schematic drawing of the hierarchical structures of the SHSSA with micro-cactus and nano-spikes, indicating the mechanisms of superhydrophobicity and selectivity (solar trapping and IR reflection). (**B**) Fabrication of the SHSSA using chemical etching of the substrate, spin coating of the TiN nanoparticles, and fluorination to render spectral selectivity as well as superhydrophobicity.

**Figure 9 nanomaterials-15-00378-f009:**
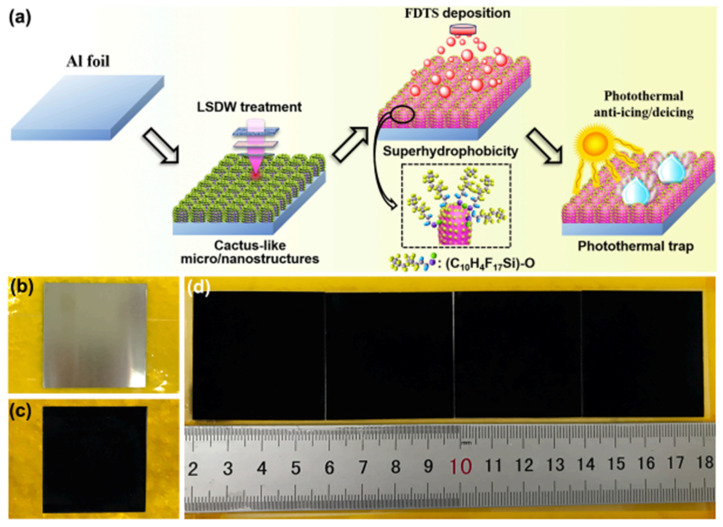
Schematic of the multifunctional Al for photothermal anti-icing [93]. (**a**) Fabrication process of the black and superhydrophobic AI surface by LSDW and TE of the FDTS protocol. (**b**) Optical photograph of the bare Al specimen; (**c**) Optical photograph of the LSDW-treated Al specimen; (**d**) Optical photograph of the large-sized multifunctional Al specimens.

**Figure 10 nanomaterials-15-00378-f010:**
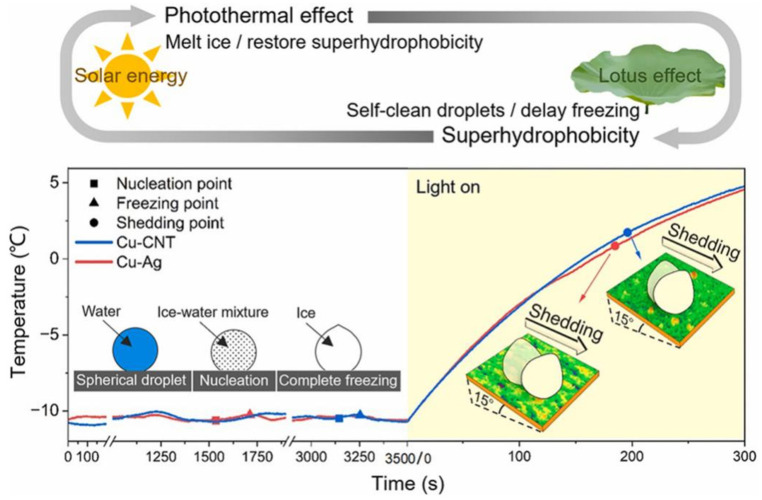
Relationship between the surface temperature of two PHS-NMSs and time [95].

**Table 1 nanomaterials-15-00378-t001:** The photothermal conversion efficiencies of photothermal nano-micro surfaces.

Material	Structure Description	Light Source	Photothermal Conversion Efficiency	Reference
NH2-MIL-125/TiN/EG Hybrid Nanofluid	Composite nanofluid with NH2-MIL-125 and TiN nanoparticles	Sunlight (UV to NIR)	83%	[23]
Gold Nanorods	Surface plasmon resonance-based nanorods	Near-Infrared (NIR)	83%	[24]
AuPt Bimetallic Nanoplates	NIR-II window excitation	NIR-II (1064 nm)	75%	[23]
Surface-Roughness-Adjustable Au Nanorods	Strong plasmon absorption and hotspots	NIR (808 nm)	78%	[23]
Multilayered Mesoporous Gold Nanoarchitecture	Multilayered nanoarchitecture for NIR control	NIR (808 nm)	85%	[23]

Note: The photothermal conversion power densities of the surfaces in this table are normalized to 1.0 (W/cm^2^).

**Table 2 nanomaterials-15-00378-t002:** Anti-icing properties of hydrophobic surfaces with nano-micro structures in ice wind tunnels.

Year	Surface Type	Key Findings	Anti-Icing Mechanism	Reference
2011	Nano-micro-structured surfaces	Hydrophobic surfaces demonstrated rapid droplet rebound and detachment under external forces, effectively reducing icing.	Hydrophobic surfaces prevent droplet adhesion, reducing ice formation.	[67]
2011	Nano-micro-structured surfaces	Energy savings of 33% and 13% under light ice and frost ice conditions, respectively.	Nano-micro structures enhance droplet detachment, reducing energy consumption for anti-icing.	[68]
2016	Different types of nano-micro-structured surfaces	Different icing processes and ice patterns formed on airfoils under the same conditions.	Surface structures influence ice nucleation and growth patterns.	[69]
2011	Hydrophobic surfaces	Water droplets on hydrophobic surfaces continuously bounced off and left the surface.	Hydrophobicity reduces droplet contact time, preventing ice formation.	[52]
2011	Hydrophobic surfaces	Water droplets on hydrophobic surfaces delayed freezing for over 2 h at −15 °C.	Hydrophobic surfaces delay ice nucleation by reducing supercooled water collection.	[60]

## Data Availability

The original contributions presented in this study are included in the article. Further inquiries can be directed to the corresponding author.

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
