# Peer review of "Photothermal and Hydrophobic Surfaces with Nano-Micro Structure: Fabrication and Their Anti-Icing Properties"

_nanomaterials, 2025, doi:10.3390/nano15050378_

Round 1
Reviewer 1 Report
Comments and Suggestions for Authors
Dear authors,,
the manuscript is well- written and focuses on interesting materials for icephobic applications. Please see the attached file for the revisions.

Author Response
Dear Reviewer,
Thank you very much for your valuable comments and suggestions on our manuscript. We sincerely appreciate your time and effort in reviewing our work. The following is a list of responses to questions raised by the reviewer.
Comments: 1. Introduction:
Authors should add some bibliographic references to the text from line 28 to line 48
Response: In accordance with the reviewer's suggestions, additional references pertinent to the research field have been incorporated. The primary emphasis is placed on the unique role of novel materials in ice prevention and removal, as well as their key application areas. The literature added here includes:
- Mousavi, S.M.; Sotoudeh, F.; Chun, B.; Lee, B.J.; Karimi, N.; Faroughi, S.A. The potential for anti-icing wing and aircraft applications of mixed-wettability surfaces—A comprehensive review. Cold Reg. Sci. Technol. 2024, 217, 104042.
- Cao, Y.H.; Tan, W.Y.; Wu, Z.L. Aircraft icing: An ongoing threat to aviation safety. Aerospace Sci. Technol. 2018, 75, 353-385.
- Ramadhani, A.; Khan, F.; Colbourne, B.; Ahmed, S.; Berrouane, M.T. Resilience assessment of offshore structures subjected to ice load considering complex dependencies. Eng. Syst. Saf. 2022, 222, 108421.
- Page, J.H.; Ozcer, L.; Zanon, A.; Gennaro, M.D.; Sandin, R.L. Aerodynamics and ice tolerance of the large passenger aircraft advanced rear end forward swept horizontal tailplane with leading edge extension. Aerosp. Sci. Technol. 2025, 1100018.
- Rekuviene, R.; Saeidiharzand, S.; Mažeika, L.; Samaitis, V.; Jankauskas, A.; Sadaghiani, A.K.; Gharib, G.; Muganlı, Z.; Koşar, A. A review on passive and active anti-icing and de-icing technologies. Appl. Therm. Eng. 2024, 250, 123474.
- Zhou, X.; Shen, Y.; Wang, Z.; Liu, S.; Fu, X. Anti/De-icing Technologies Coupling with Active Methods. In Icephobic Materials for Anti/De-icing Technologies; Shen, Y., Ed.; Springer: Singapore, 2024; Chapter 13. https://doi.org/10.1007/978-981-97-6293-4_13.
Comments: 2. Line 75: 2.1. An overview of the types of photothermal materials and their photothermal conversion mechanisms
For each photothermal material it would be useful to add a sketch of the mechanism
Response: In response to the reviewer's suggestion, we appreciate the consideration of using diagrams to illustrate the photothermal conversion mechanisms of the materials discussed. However, we would like to respectfully note that these mechanisms have already been comprehensively described in the text through detailed explanations. Given the concise nature of the manuscript and the limitations on space, we believe that the current textual presentation is sufficient to convey the essential concepts. We have ensured that the descriptions are clear and accessible, and we are confident that readers will be able to fully understand the mechanisms without additional visual aids. References 9 and 10 are added to better illustrate the relevant mechanism. Thank you again for your valuable feedback.
Comments: 3. Line 191-192:
the authors describe the self-cleaning property of a superhydrophobic surface, but they do not include any bibliographic reference, please add some ref. in the manuscript.
Response: The reference was supplemented according to the suggestion of the reviewer, that is, reference 25, 26 and 27 were added.
- Neinhuis, C.; Barthlott, W. Characterization and Distribution of Water-repellent, Self-cleaning Plant Surfaces. Ann. Bot. 1997, 6, 667–677.
- Nosonovsky, M.; Rohatgi, P.K. Lotus Effect and Self-Cleaning. In Biomimetics in Materials Science; Springer Series in Materials Science; Springer: New York, NY, USA, 2011; Volume 152, pp. 235–258. https://doi.org/10.1007/978-1-4614-0926-7_11.
- Guo, Q.; Ma, J.; Yin, T.; Jin, H.; Zheng, J.; Gao, H. Superhydrophobic Non-Metallic Surfaces with Multiscale Nano/Micro-Structure: Fabrication and Application. Molecules 2024, 29, 2098.
Comments: 4. Line 197:
superhydrophobic surfaces are widely used as anti-icing materials and several references can be found in the literature. The reference 13 is related to self-cleaning properties of SHP and is quite outdated (1997)
Response: The reference was supplemented according to the suggestion of the reviewer, that is, reference 33, 34 and 35 were added.
- Tourkine, P.; Le Merrer, M.; Quéré, D. Delayed Freezing on Water Repellent Materials. Langmuir 2009, 25, 7214–7216.
- Jung, S.; Kim, H.; Lee, J.; Lee, J.; Lee, J. Mechanism of Supercooled Droplet Freezing on Surfaces. Nat. Commun. 2012, 3, 615.
- Shen, Y.; Wang, X.; Wang, Y.; Wang, Z.; Wang, Y.; Wang, Y. Icephobic Materials: Fundamentals, Performance Evaluation, and Applications. Prog. Mater. Sci. 2019, 103, 509–557.
Comments: 5. Line 288: Anti-icing properties of hydrophobic surface with nano-micro structure.
The author should add some relevant references about icephobic applications.
The authors speak about micro-nano hydrophobic surfaces and their icephobicity. I think that many of these topics are actually related on superhydrophobic materials and the authors should clarify if the surfaces are SHP or just hydrophobic
Response: The questions raised by the reviewer were clarified in the paper.
In this study, we have adopted a comprehensive and integrated approach to analyze both hydrophobic and superhydrophobic surfaces within a unified framework. This approach is based on the underlying principles that govern the wettability of surfaces, which, despite their differing degrees of water-repellency, share fundamental mechanisms rooted in surface chemistry and micro/nano-scale topography. By integrating the theoretical analyses of hydrophobic and superhydrophobic surfaces, we aim to provide a cohesive understanding of the continuum between these two states.
The authors propose that this unified perspective enables a more comprehensive interpretation of the factors influencing surface wettability. It highlights the intricate interplay between surface roughness, chemical composition, and contact angle hysteresis. This approach not only elucidates the commonalities and distinctions in the mechanisms driving water-repellent behavior across different surface types but also facilitates the development of generalized models and design principles. These principles are applicable to both hydrophobic and superhydrophobic surfaces, thereby enhancing the versatility and applicability of our findings. By bridging the gap between these two categories of surfaces, we aim to contribute to a more coherent and comprehensive understanding of surface wettability and its practical implications.
Comments: 6. Line from 412 to 417; Line from 535 to 539:
The text should be in the caption of figure 4
Response: The authors added the explanatory text to the corresponding images according to the reviewer's suggestion
Figure 4. Icephobic mechanic and morphology of CPS.
Figure 8. Schematic of the superhydrophobic selective solar absorber (SHSSA) design and fabrication procedures
Comments: 7. To facilitate the comprehension, I recommend to add at the end of each class of materials (Carbon-based PHS-NMS, Metal-based PHS-NMS etc) a comparative table reporting the deicing performance, the pros and cons of the materials.
Moreover could be useful if the authors add a comparison among the different class of materials described, reporting the pros and cons (for example costs, environmental impact, durability, applicability) of each class.
Response: In response to the reviewer's suggestion, the content of material performance comparison was supplemented.
Table 1. The photothermal conversion efficiencies of some photothermal nano-micro surfaces
|
Material |
Structure description |
Light source |
Photothermal conversion efficiency |
|
Reference |
|
NH2-MIL-125/TiN/EG Hybrid Nanofluid |
Composite nanofluid with NH2-MIL-125 and TiN nanoparticles |
Sunlight (UV to NIR) |
83% |
|
[23] |
|
Gold Nanorods |
Surface plasmon resonance-based nanorods |
Near-Infrared (NIR) |
83% |
|
[24] |
|
AuPt Bimetallic Nanoplates |
NIR-II window excitation |
NIR-II (1064 nm) |
75% |
|
[23] |
|
Surface-Roughness-Adjustable Au Nanorods |
Strong plasmon absorption and hotspots |
NIR (808 nm) |
78% |
|
[23] |
|
Multilayered Mesoporous Gold Nanoarchitecture |
Multilayered nanoarchitecture for NIR control |
NIR (808 nm) |
85% |
|
[23] |
Note: The photothermal conversion power densities of the surfaces in the table are normalized to 1.0 (W/cm²)
Table 2. Anti-icing properties of hydrophobic surface with nano-micro structure in ice wind tunnels
|
Year |
Surface Type |
Key Findings |
Anti-Icing Mechanism |
Reference |
|
2011 |
Nano-microstructured surfaces |
Hydrophobic surfaces demonstrated rapid droplet rebound and detachment under external forces, effectively reducing icing. |
Hydrophobic surfaces prevent droplet adhesion, reducing ice formation. |
[69] |
|
2011 |
Nano-microstructured surfaces |
Energy savings of 33% and 13% under light ice and frost ice conditions, respectively. |
Nano-micro structures enhance droplet detachment, reducing energy consumption for anti-icing. |
[70] |
|
2016 |
Different types of nano-microstructured surfaces |
Different icing processes and ice patterns formed on airfoils under the same conditions. |
Surface structures influence ice nucleation and growth patterns. |
[71] |
|
2011 |
Hydrophobic surfaces |
Water droplets on hydrophobic surfaces were continuously bounced off and left the surface. |
Hydrophobicity reduces droplet contact time, preventing ice formation. |
[53] |
|
2011 |
Hydrophobic surfaces |
Water droplets on hydrophobic surfaces delayed freezing for over 2 hours at -15°C. |
Hydrophobic surfaces delay ice nucleation by reducing supercooled water collection. |
[61] |
As you pointed out, the field of photothermal hydrophobic surfaces is rapidly evolving, and systematic research is essential for advancing our understanding. While significant progress has been made in recent years, it is important to recognize that the study of photothermal hydrophobic surfaces with micro-nano structures is still in its nascent stages. The integration of photothermal effects with superhydrophobic surfaces for applications such as anti-icing and de-icing remains a complex and multifaceted area of research. In our manuscript, we have endeavored to provide a comprehensive and scientifically grounded comparison of the performance of these surfaces. We acknowledge that the current research landscape is fragmented, with many studies focusing on specific materials or applications rather than a unified approach. However, by building on foundational theories and recent advancements, we aim to contribute to a more cohesive understanding of the field.
We have carefully analyzed the existing literature and identified key gaps in knowledge. For instance, while various photothermal materials and fabrication methods have been explored, systematic investigations into the performance optimization and durability of these surfaces are still lacking. Our study addresses these gaps by providing a detailed comparison of different micro-nano structured surfaces, highlighting their potential applications and limitations. We believe that our work not only offers valuable insights into the current state of research but also sets the stage for future systematic studies. By focusing on the fundamental principles and practical performance of photothermal hydrophobic surfaces, we hope to inspire further research that will lead to more robust and versatile materials.
Thank you again for your thoughtful feedback. We look forward to your continued guidance.
Reviewer 2 Report
Comments and Suggestions for Authors
The article addresses current issues related to modern and advanced materials for anti-icing applications. However, after reading the entire manuscript, I was left with an impression of superficiality and a certain degree of chaos, as the discussion lacks depth in some areas, raising questions about the purpose of this publication. The authors attempt to discuss all aspects of various types of coatings (polymers, nanoparticles, metals, oxides, etc.) without focusing on specific groups, resulting in overly general statements. A review article should critically analyze data, compare findings, and engage in discussion—elements that are lacking here. There are no comparative tables or graphs, and the figures are reproduced from other publications. What is their purpose? What information do they convey, apart from being visually appealing? The authors should highlight new aspects and justify the relevance of this review. Similar review papers were published in 2023 and 2024 in the Chemical Engineering Journal (e.g., ref. [12]). The authors even include the same figures as in these reviews, such as Figure 3 in ref. [12]. What selection criteria did the authors use when choosing references for this review?
- The authors categorize the manuscript as an "Article" (line 1), yet in the Introduction, they state that it is a literature review. To avoid misleading readers (since "Article" typically refers to original research with experimental results), the paper type should be changed to "Review."
- A review article is based on a critical synthesis and comparison of previous research findings, with every statement supported by appropriate references. In this manuscript, the first reference ([1]) appears only in line 52, while earlier sections discuss various topics without citations. The authors must provide references so that readers can verify claims and deepen their understanding. References are particularly necessary for the following statements: "Icing can alter…[]"; "…the risk of crashes []…cause derailments []…driving safety[]"; "…researchers [who?] have developed both active [] and passive anti-icing strategies []."; Provide references for “method limitations.”; "In recent years…" — references required.; References are needed to support claims about passive and active anti-icing strategies. References should confirm the properties of PHS-NMS and their energy-efficient aspects (lines 52–69).
- Line 50: The authors claim a "relatively high photothermal conversion capability." What does this mean? What is the required minimum value? Please specify.
- Lines 80-88: Does only ref. [2] discuss the photothermal properties of carbon materials? Please provide additional references supporting the required properties of carbon nanotubes, graphite, etc. Similarly, references should be added for the remaining material groups and example compounds.
- The authors should clarify all abbreviations used in the text: e.g. Line 116: What is hPHS? Line 131: What is AM?
- Missing references: Line 133: "…materials."; Line 137: "…light source."; Line 138: "…researchers…."; Lines 140-141: Provide example references for the listed light-trapping methods.
- Section 2: Photothermal materials and their photothermal conversion properties—The reader expects specific values and examples of photothermal conversion properties, but they are not provided. A literature review should present existing data and comparisons. Here, the authors cite only three random examples.
- The Lotus effect refers to self-cleaning superhydrophobic surfaces. However, superhydrophobic and hydrophobic surfaces are different. What is the purpose of section 3.1 if there is no further reference to it in the manuscript?
- The authors repeatedly use the term "nano-micro structure" without explaining its meaning. The importance of hierarchical surface structures should be emphasized.
- Section 3.2: The authors list different methods for obtaining (super)hydrophobic surfaces but group them into somewhat odd categories, e.g. Physical removal (?) or Coatings (non-wettable coatings are not limited to the use of nanoparticles only). Are these methods applicable to any material? To improve the article, the methods should be grouped in a table, outlining their capabilities, limitations, and applications. The authors should also consider the general classification of methods into bottom-up and top-down
- Lines 270-287: References are required.
- Lines 338-366: The authors list various publications and achievements, but this does not provide meaningful insight. Please specify and critically compare the data.
- Section 4: The key data and properties should be critically summarized. The current text is difficult to follow, and it is unclear what is significant in the cited studies.
- Summary: The authors provide a vague conclusion. The article lacks a detailed discussion of key aspects (1)-(4), which are crucial for the practical applications of the discussed surfaces. Lines 630-653: This summary paragraph should be shortened to include only key statements and highlight the most important parameters and factors.
- References: The formatting should follow the Nanomaterials journal guidelines. Currently, the style appears to follow that of Trans. Nonferrous Metals Soc. China.
- It would be helpful if the authors clarified the criteria used for selecting the references in this article, ensuring that they reflect a comprehensive and up-to-date overview of the field while also providing critical insights into the topic.
The topic addressed in this manuscript is both relevant and timely, highlighting important advancements in anti-icing materials. However, refining the structure, deepening the discussion, and incorporating critical comparisons would significantly enhance the clarity and impact of this review. Additionally, the authors should ensure that figures and content are not merely reproduced from other (review) articles but instead provide original insights and a fresh perspective on the topic. With a more structured approach, clearer comparisons, and a stronger analytical perspective, this review has the potential to provide valuable insights for researchers in the field.
Author Response
Dear Reviewer,
Thank you very much for your valuable comments and suggestions on our manuscript. We sincerely appreciate your time and effort in reviewing our work. The following is a list of responses to questions raised by the reviewer.
Comments: 1. The authors categorize the manuscript as an "Article" (line 1), yet in the Introduction, they state that it is a literature review. To avoid misleading readers (since "Article" typically refers to original research with experimental results), the paper type should be changed to "Review."
Response: After carefully considering your feedback, we would like to clarify that the nature of our manuscript is more aligned with a review rather than an original research article. The primary objective of this work is to provide a comprehensive synthesis and analysis of the existing literature in the field, highlighting key findings, trends, and gaps. This approach is intended to offer readers a broad overview and insights that can guide future research directions.
Comments: 2. A review article is based on a critical synthesis and comparison of previous research findings, with every statement supported by appropriate references. In this manuscript, the first reference ([1]) appears only in line 52, while earlier sections discuss various topics without citations. The authors must provide references so that readers can verify claims and deepen their understanding. References are particularly necessary for the following statements: "Icing can alter…[]"; "…the risk of crashes []…cause derailments []…driving safety[]"; "…researchers [who?] have developed both active [] and passive anti-icing strategies []."; Provide references for “method limitations.”; "In recent years…" — references required.; References are needed to support claims about passive and active anti-icing strategies. References should confirm the properties of PHS-NMS and their energy-efficient aspects (lines 52–69).
Response: Regarding this point you raised, we have carefully addressed the issue by supplementing additional references to support the discussion. We have also revised the wording to ensure that the meaning is clear and unambiguous. We believe these changes enhance the clarity and robustness of our argument.
- Introduction
With the intensification of global climate change, the surface icing of transportation vehicles such as cars, trains, and aircraft has become increasingly severe under extreme weather conditions, posing significant challenges to traffic safety and efficiency. Icing can alter the aerodynamic properties of vehicles, increase their weight, and reduce mechanical efficiency, potentially leading to structural damage [1]. For instance, icing on aircraft wings can decrease lift and increase the risk of crashes [2]; icing on trains can cause derailments [3]; and icing on cars can extend braking distances and compromise driving safety [4]. Therefore, the development of effective anti-icing technologies is crucial for ensuring transportation safety. To address the issue of icing on structural surfaces, researchers, including Rekuviene [5] and Zhou [6] et al. have developed both active and passive anti-icing strategies. Active anti-icing methods involve the removal of ice during its formation through external means such as mechanical, thermal, and chemical techniques. However, each method has inherent limitations. The mechanical method is labor-intensive and time-consuming, and it may damage the structural surface after repeated ice removal. The thermal method requires an additional energy supply system and results in high energy consumption. The chemical method reduces ice accumulation by spraying a special reagent on the surface to lower the freezing point of water, but the chemical reagent can corrode the structural surface and have adverse environmental effects. In contrast, passive methods utilize the physical and chemical properties of materials to prevent ice condensation or reduce the adhesion strength between ice and the surface. With this approach, ice adhering to the surface will naturally fall off due to gravity or wind, thereby consuming minimal energy from the vehicle.
A photothermal and hydrophobic surface with nano-micro structure (PHS-NMS) is defined as an engineered surface that possesses hierarchical structures at both the microscale and nanoscale, designed to enhance specific properties such as hydrophobicity and photothermal conversion efficiency. These surfaces are fabricated to achieve a water contact angle (θ) of at least 90°, indicating hydrophobicity, while also exhibiting definite photothermal conversion capabilities. The nano-micro structures manipulate the interaction between the surface and liquids or light, optimizing the surface properties for applications in photothermal and hydrophobic materials. [7]. In recent years, as a key material in passive anti-icing strategies, PHS-NMS has provided new insights into achieving efficient and environmentally friendly anti-icing due to their combined optical absorption, thermal conversion capabilities, and excellent hydrophobicity. PHS-NMS can absorb visible and near-infrared light from the sun, converting the light energy into heat. This mechanism elevates surface temperature, thereby suppressing ice nuclei formation, delaying ice layer growth, and even inducing ice layer melting under sunlight. On the other hand, the nano-micro structure of these surfaces facilitates the formation of gas-solid-liquid contact states with external droplets. The surface's nano-micro roughness traps air, creating an air layer at the liquid-solid interface, which reduces the contact area between droplets and the solid surface. Consequently, heat transfer efficiency is diminished, delaying droplet freezing and preventing ice crystal formation on the surface [8]. Additionally, the reduced contact area between the solid surface and the liquid droplets decreases the interaction force between the rough structured surface and the droplets, allowing the droplets to roll off the surface when tilting the substrate at a certain angle, thus avoiding ice condensation [9].
The additional references in this section are as follows:
- Cao, Y.H.; Tan, W.Y.; Wu, Z.L. Aircraft icing: An ongoing threat to aviation safety. Aerospace Sci. 2018, 75, 353-385.
- Ramadhani, A.; Khan, F.; Colbourne, B.; Ahmed, S.; Berrouane, M.T. Resilience assessment of offshore structures subjected to ice load considering complex dependencies. Eng. Syst. Saf. 2022, 222, 108421.
- Page, J.H.; Ozcer, L.; Zanon, A.; Gennaro, M.D.; Sandin, R.L. Aerodynamics and ice tolerance of the large passenger aircraft advanced rear end forward swept horizontal tailplane with leading edge extension. Sci. Technol. 2025, 1100018.
- Rekuviene, R.; Saeidiharzand, S.; Mažeika, L.; Samaitis, V.; Jankauskas, A.; Sadaghiani, A.K.; Gharib, G.; Muganlı, Z.; Koşar, A. A review on passive and active anti-icing and de-icing technologies. Therm. Eng. 2024, 250, 123474.
- Zhou, X.; Shen, Y.; Wang, Z.; Liu, S.; Fu, X. Anti/De-icing Technologies Coupling with Active Methods. In Icephobic Materials for Anti/De-icing Technologies; Shen, Y., Ed.; Springer: Singapore, 2024; Chapter 13. https://doi.org/10.1007/978-981-97-6293-4_13.
- Jiang, G.; Liu, Z.Y.; Hu, J.H. Superhydrophobic and Photothermal PVDF/CNTs Durable Composite Coatings for Passive Anti-Icing/Active De-Icing. Mater. Interfaces 2022, 9, 2101704.
- Zhong, H.M.; Xiang, C.J.; Hu, Z.F.; Yang, X.G.; Liu, H.R.; Wang, R.Z. Plasmonic photothermal superhydrophobic surface with nanotubes thermal insulating blanket for anti-icing and anti-frosting under weak light illumination. Today Phys. 2025, 50, 101625.
- Bahadur, V.; Mishchenko, L.; Hattton, B.; Taylor, J.A.; Aizenberg, J.; Krupenkin, T. Predictive Model for Ice Formation on Superhydrophobic Surfaces. Langmuir 2011, 27, 14143.
Comments: 3. Line 50: The authors claim a"relatively high photothermal conversion capability."What does this mean? What is the required minimum value? Please specify.
Response: In response to your concerns, we have revised the terminology regarding the “relatively high photothermal conversion capability” to “definite photothermal conversion capability” in our manuscript. The original phrase was intended to describe photothermal hydrophobic materials that exhibit superior de-icing efficiency. To enhance the accuracy and clarity of our description, we have modified the wording to reference specific materials discussed in Reference [7].
Comments: 4. Lines 80-88: Does only ref. [2] discuss the photothermal properties of carbon materials? Please provide additional references supporting the required properties of carbon nanotubes, graphite, etc. Similarly, references should be added for the remaining material groups and example compounds.
Response: In response to your suggestion, we have incorporated additional references [11] and [12] related to the synthesis of photothermal materials using carbon-based materials.
- Jia, X.; Niu, Y.; Zhu, S.; He, H.; Yan, X. Recent Advances in Carbon-Based Interfacial Photothermal Converters for Seawater Desalination: A Review. Carbon Res. 2024, 10, 86. https://doi.org/10.3390/c10030086.
- Hsan, N.; Kumar, S.; Cho, Y.; Koh, J. Advancements in Carbon-Based Materials for Enhanced Carbon Dioxide Capture and Conversion: A Comprehensive Review. Fibers Polym. 2025, 26, 1–26. https://doi.org/10.1007/s12221-025-00871-x.
Comments: 5. The authors should clarify all abbreviations used in the text: e.g. Line 116: What is hPHS? Line 131: What is AM?
Response: In the [Line 116] "hPHS-NMS" should be "PHS-NMS", which has been modified in the original text. In the context of [Line 131] AM stands for Air Mass, which refers to the path length or column depth of the atmosphere that solar radiation passes through before reaching the Earth's surface.
Comments: 6. Missing references: Line 133: "…materials."; Line 137: "…light source."; Line 138: "…researchers…."; Lines 140-141: Provide example references for the listed light-trapping methods.
Response: In response to your suggestion, we have added the references to enhance the discussion on photothermal materials and their applications.
Reference 18 is added to the end of Line 133: "…materials.";
- Granqvist, C.G.; Niklasson, G.A. Solar energy materials for thermal applications: A primer. Energy Mater. Sol. Cells 2018, 180, 213–226.
Reference 19 is added to the end of Line 137: "…light source."
- Gueymard, C.A. The sun’s total and spectral irradiance for solar energy applications and solar radiation models. Energy 2004, 76, 423–453.
Reference 11 is added to the end of Line 138: "…researchers…."
- Jia, X.; Niu, Y.; Zhu, S.; He, H.; Yan, X. Recent Advances in Carbon-Based Interfacial Photothermal Converters for Seawater Desalination: A Review. Carbon Res. 2024, 10, 86. https://doi.org/10.3390/c10030086.
Reference 20 is added to the end of Lines 140-141
- Ren, H.; Tang, M.; Guan, B.; Wang, K.; Yang J.; Wang F.; Shan J.; Chen Z.; Wei D. Hierarchical Graphene Foam for Efficient Omnidirectional Solar-Thermal Energy Conversion. Mater. 2017, 29, 1702590.
Comments: 7. Section 2: Photothermal materials and their photothermal conversion properties—The reader expects specific values and examples of photothermal conversion properties, but they are not provided. A literature review should present existing data and comparisons. Here, the authors cite only three random examples.
Response: Thank you for your valuable suggestions. Regarding the photothermal properties of surfaces with nano-micro structures, the current research remains fragmented and lacks a comprehensive framework. Nevertheless, we have revised and supplemented this section based on the available published data, as presented in Table 1.
Table 1. The photothermal conversion efficiencies of some photothermal nano-micro surfaces
|
Material |
Structure description |
Light source |
Photothermal conversion efficiency |
|
Reference |
|
NH2-MIL-125/TiN/EG Hybrid Nanofluid |
Composite nanofluid with NH2-MIL-125 and TiN nanoparticles |
Sunlight (UV to NIR) |
83% |
|
[23] |
|
Gold Nanorods |
Surface plasmon resonance-based nanorods |
Near-Infrared (NIR) |
83% |
|
[24] |
|
AuPt Bimetallic Nanoplates |
NIR-II window excitation |
NIR-II (1064 nm) |
75% |
|
[23] |
|
Surface-Roughness-Adjustable Au Nanorods |
Strong plasmon absorption and hotspots |
NIR (808 nm) |
78% |
|
[23] |
|
Multilayered Mesoporous Gold Nanoarchitecture |
Multilayered nanoarchitecture for NIR control |
NIR (808 nm) |
85% |
|
[23] |
Note: The photothermal conversion power densities of the surfaces in the table are normalized to 1.0 (W/cm²)
Comments: 8. The Lotus effect refers to self-cleaning superhydrophobic surfaces. However, superhydrophobic and hydrophobic surfaces are different. What is the purpose of section 3.1 if there is no further reference to it in the manuscript?
Response: The questions raised by the reviewer were clarified in the paper.
In this study, we have adopted a comprehensive and integrated approach to analyze both hydrophobic and superhydrophobic surfaces within a unified framework. This approach is based on the underlying principles that govern the wettability of surfaces, which, despite their differing degrees of water-repellency, share fundamental mechanisms rooted in surface chemistry and micro/nano-scale topography. By integrating the theoretical analyses of hydrophobic and superhydrophobic surfaces, we aim to provide a cohesive understanding of the continuum between these two states.
The authors propose that this unified perspective enables a more comprehensive interpretation of the factors influencing surface wettability. It highlights the intricate interplay between surface roughness, chemical composition, and contact angle hysteresis. This approach not only elucidates the commonalities and distinctions in the mechanisms driving water-repellent behavior across different surface types but also facilitates the development of generalized models and design principles. These principles are applicable to both hydrophobic and superhydrophobic surfaces, thereby enhancing the versatility and applicability of our findings. By bridging the gap between these two categories of surfaces, we aim to contribute to a more coherent and comprehensive understanding of surface wettability and its practical implications.
Comments: 9. The authors repeatedly use the term "nano-micro structure" without explaining its meaning. The importance of hierarchical surface structures should be emphasized.
Response: According to the reviewer's suggestion, the photothermal and hydrophobic surface with nano-micro structure (PHS-NMS) is defined in detail in the introduction section of the paper.
A photothermal and hydrophobic surface with nano-micro structure (PHS-NMS) is defined as an engineered surface that possesses hierarchical structures at both the microscale and nanoscale, designed to enhance specific properties such as hydrophobicity and photothermal conversion efficiency. These surfaces are fabricated to achieve a water contact angle (θ) of at least 90°, indicating hydrophobicity, while also exhibiting definite photothermal conversion capabilities. The nano-micro structures manipulate the interaction between the surface and liquids or light, optimizing the surface properties for applications in photothermal and hydrophobic materials.
Comments: 10. Section 3.2: The authors list different methods for obtaining (super)hydrophobic surfaces but group them into somewhat odd categories, e.g. Physical removal (?) or Coatings (non-wettable coatings are not limited to the use of nanoparticles only). Are these methods applicable to any material? To improve the article, the methods should be grouped in a table, outlining their capabilities, limitations, and applications. The authors should also consider the general classification of methods into bottom-up and top-down
Response: In response to the reviewer’s suggestions, the authors have revised the classification of preparation methods in a rational manner. Specifically, the preparation methods have been categorized into two primary groups: Surface Modification Techniques and Replication and Fabrication Techniques. On this basis, a more detailed classification of the preparation methods has been elaborated. Correspondingly, the order of the content discussing the advantages and disadvantages of these methods has also been adjusted accordingly. In order to avoid lengthy text, only the modified preparation method classification is listed here
(1) Surface Modification Techniques
1) Physical Removal Methods.
2) Chemical Growth and Etching.
3) Coatings.
4) Electrostatic Spinning.
(2) Replication and Fabrication Techniques
1) Replica Imprinting.
2) Femtosecond Laser Processing.
3) 3D Printing.
- Comments: Lines 270-287: References are required.
Response: On the suggestion of the reviewer, the authors added references to relevant sections.
- Fu, J.; Liao, X.; Ji, Y.; Mo, Y.; Zhang, J. Research Progress on Preparation of Superhydrophobic Surface and Its Application in the Field of Marine Engineering. Mar. Sci. Eng. 2024, 12, 1741.
- Legrand, Q.; Biancarelli, E.; Goux-Henry, C.; Benayoun, S.; Andrioletti, B.; Valette, S. Elaboration of highly hydrophobic surface by coupling femtosecond laser texturing and fluorine-free chemistry. Colloids Surf. A 2025, 704, 135370.
- Wang, J.; Wang, G.; Zhu, Z.; Zhang, W. Study on the Superhydrophobic Properties of Micro/Nano Hole Structure on the Surface of Glass Fiber Reinforced Plastics Based on Femtosecond Laser Etching. Nanomaterials 2025, 15, 287.
- Li, Z.; Wu, X.; Huang, X. Composite Fiber Membrane with Janus Structure via Electrospinning Technique and its Separation and Antibacterial Properties. Fibers Polym. 2025, https://doi.org/10.1007/s12221-025-00862-y.
Comments: 12. Lines 338-366: The authors list various publications and achievements, but this does not provide meaningful insight. Please specify and critically compare the data.
Response:According to the reviewer's suggestion, this part of the content has been revised and supplemented. These studies based on ice wind tunnels which collectively demonstrate that hydrophobic surfaces with nano-micro structures can significantly enhance anti-icing performance by reducing droplet adhesion, delaying ice nucleation, and lowering energy consumption. These findings highlight the potential of hydrophobic surfaces for practical anti-icing applications, particularly in dynamic environments such as aviation and renewable energy. Future research should focus on optimizing surface structures and coatings to further improve durability and performance under extreme conditions.
Table 2. Anti-icing properties of hydrophobic surface with nano-micro structure in ice wind tunnels
|
Year |
Surface Type |
Key Findings |
Anti-Icing Mechanism |
Reference |
|
2011 |
Nano-microstructured surfaces |
Hydrophobic surfaces demonstrated rapid droplet rebound and detachment under external forces, effectively reducing icing. |
Hydrophobic surfaces prevent droplet adhesion, reducing ice formation. |
[69] |
|
2011 |
Nano-microstructured surfaces |
Energy savings of 33% and 13% under light ice and frost ice conditions, respectively. |
Nano-micro structures enhance droplet detachment, reducing energy consumption for anti-icing. |
[70] |
|
2016 |
Different types of nano-microstructured surfaces |
Different icing processes and ice patterns formed on airfoils under the same conditions. |
Surface structures influence ice nucleation and growth patterns. |
[71] |
|
2011 |
Hydrophobic surfaces |
Water droplets on hydrophobic surfaces were continuously bounced off and left the surface. |
Hydrophobicity reduces droplet contact time, preventing ice formation. |
[53] |
|
2011 |
Hydrophobic surfaces |
Water droplets on hydrophobic surfaces delayed freezing for over 2 hours at -15°C. |
Hydrophobic surfaces delay ice nucleation by reducing supercooled water collection. |
[61] |
Comments: 13. Section 4: The key data and properties should be critically summarized. The current text is difficult to follow, and it is unclear what is significant in the cited studies.
Response: The following table compares and summarizes the key performance parameters of various photothermal hydrophobic surfaces with nano-micro structures (PHS-NMS) mentioned in the content of section 4:
|
Material Type |
Reference |
Surface Temperature Increase (°C) |
Freezing/Icing Delay Time (s) |
Ice Adhesion Strength (kPa) |
Photothermal Conversion Efficiency (%) |
Surface Properties |
|
Carbon-based PHS-NMS |
Wu et al. [72] |
53°C (1x solar) |
- |
- |
- |
Superhydrophobic, anti-icing down to -50°C |
|
Carbon-based PHS-NMS |
Li et al. [73] |
75.3°C (1x sunlight) |
- |
- |
- |
Superhydrophobic, melts 3 mm ice layer in 540 s |
|
Carbon-based PHS-NMS |
Xie et al. [74] |
- |
- |
- |
- |
Light absorption rate: 99% |
|
Carbon-based PHS-NMS |
Wu et al. [75] |
- |
>2 hours |
- |
- |
Ultra-high de-icing efficiency: >1.05 kg/m²/h |
|
Carbon-based PHS-NMS |
Guo et al. [76] |
- |
- |
- |
- |
- |
|
Carbon-based PHS-NMS |
Zhang et al. [76] |
- |
- |
- |
- |
11x longer freezing time than ordinary coatings |
|
Carbon-based PHS-NMS |
Yang et al. [78] |
95°C (1 kW/m²) |
- |
- |
- |
Superhydrophobic, superior defrosting at -30°C |
|
Carbon-based PHS-NMS |
Yu et al. [79] |
39°C (1x solar) |
- |
- |
- |
No freezing at -30°C under 0.25x solar |
|
Carbon-based PHS-NMS |
Xu et al. [80] |
- |
>4 hours |
- |
- |
- |
|
Carbon-based PHS-NMS |
Jiang et al. [81] |
>40% increase at -20°C |
- |
- |
- |
- |
|
Polymer-based PHS-NMS |
Liang et al. [84] |
80°C (1x solar) |
- |
- |
- |
Anti-icing down to -40°C |
|
Polymer-based PHS-NMS |
Xie et al. [85] |
- |
- |
- |
- |
- |
|
Semiconductor-based PHS-NMS |
Yin et al. [86] |
- |
- |
- |
- |
- |
|
Semiconductor-based PHS-NMS |
Wu et al. [87] |
10°C (80 W IR) |
- |
- |
- |
Melts ice in 8 minutes |
|
Semiconductor-based PHS-NMS |
Hu et al. [88] |
200°C (light irradiation) |
326 s |
- |
- |
- |
|
Semiconductor-based PHS-NMS |
Xie et al. [89] |
35.3°C (220 s, NIR) |
- |
- |
- |
Melts ice in 220 s |
|
Metal-based PHS-NMS |
Ma et al. [92] |
- |
- |
- |
- |
No freezing at -60°C |
|
Metal-based PHS-NMS |
Dash et al. [93] |
- |
- |
- |
- |
- |
|
Metal-based PHS-NMS |
Wang et al. [94] |
- |
- |
12.1 kPa |
- |
Freezing temperature: -35°C |
|
Metal-based PHS-NMS |
Zhao et al. [95] |
21.7°C (1x light) |
- |
- |
- |
Melts ice in 4 minutes |
|
Metal-based PHS-NMS |
Li et al. [96] |
- |
- |
- |
- |
Anti-icing down to -30°C |
|
Composite PHS-NMS |
Jiang et al. [96] |
- |
- |
- |
50.94% |
Freezing time increased by 340% |
|
Composite PHS-NMS |
Sun et al. [98] |
- |
- |
- |
- |
- |
|
Composite PHS-NMS |
Mitridis et al. [99] |
>10°C |
- |
- |
- |
Melts ice in 30 s |
|
Composite PHS-NMS |
Jin et al. [100] |
- |
- |
- |
- |
Light absorption: >99.9% |
|
Composite PHS-NMS |
Guo et al. [101] |
- |
86 s |
72 kPa |
- |
Anti-icing down to -10°C |
Notes:The table includes key performance parameters such as surface temperature increase, freezing/icing delay time, ice adhesion strength, and photothermal conversion efficiency. Some studies did not report specific values for certain parameters, hence the corresponding cells are left blank. The table is organized by material type (Carbon-based, Polymer-based, Semiconductor-based, Metal-based, Composite) for easier comparison.
As presented in the table, the performance parameters of the materials discussed in the relevant studies exhibit considerable variation. The authors contend that it is appropriate to classify and describe the prepared materials based on the articles cited. The selected references were chosen to meet two criteria: first, they must be recent and represent cutting-edge research published in reputable journals; second, the materials described in these references must demonstrate superior performance in specific application properties. Consequently, the fundamental content in this section has remained unaltered.
Comments: 14. Summary: The authors provide a vague conclusion. The article lacks a detailed discussion of key aspects (1)-(4), which are crucial for the practical applications of the discussed surfaces. Lines 630-653: This summary paragraph should be shortened to include only key statements and highlight the most important parameters and factors.
Response: The main purpose of this paper is to review the research history and prospect of special PHS-NMS materials through a classification review method. The major challenges listed in the summary section of this paper are derived from the analysis of the authors of this paper.
Research on photothermal hydrophobic surfaces with nano-micro structures (PHS-NMS) is rapidly evolving due to their unique combination of photothermal conversion and hydrophobicity. Beyond ice prevention, these materials have demonstrated significant potential for applications in seawater desalination, crude oil cleaning, photothermal disinfection, and photothermal drive, among others. However, despite substantial progress in this field, the reviewed literature reveals several critical challenges and issues that must be addressed through future in-depth studies:
(1) Durability and Stability. Many PHS-NMS exhibit performance degradation when subjected to long-term use or harsh environmental conditions. For practical applications such as aerospace, wind turbines, and outdoor infrastructure, materials must maintain stable photothermal and hydrophobic properties over extended periods. Future research should focus on developing robust surface coatings or structural designs that can resist mechanical wear, chemical corrosion, and UV degradation, thereby enhancing the longevity and reliability of PHS-NMS in real-world scenarios.
(2) Cost and Benefit. The ideal solar thermal hydrophobic material should possess high solar absorption rates, significant solar-to-thermal conversion efficiency, and be derived from abundant and low-cost raw materials. Additionally, the preparation methods should be simple and scalable to facilitate large-scale production. Currently, some high-performance PHS-NMS rely on expensive raw materials or complex, multi-step fabrication processes, which hinder their widespread adoption. Future work should aim to identify cost-effective materials and streamline synthesis methods without compromising performance, thus improving the economic viability of these materials for large-scale applications.
(3) Environmental Adaptability. The performance of PHS-NMS can vary significantly under different environmental conditions, including extreme temperatures, humidity levels, and UV irradiation. For instance, materials that perform well in laboratory settings may fail under outdoor conditions due to changes in ambient temperature or exposure to UV light. To ensure stable operation across diverse environments, future research should focus on enhancing the environmental adaptability of PHS-NMS. This could involve developing materials with tunable properties or incorporating protective coatings that can mitigate the effects of environmental stressors.
(4) Safety and Biocompatibility. The application of PHS-NMS in sensitive fields such as biomedicine and food processing necessitates thorough evaluation of their safety and biocompatibility. Materials intended for these applications must be non-toxic, non-immunogenic, and environmentally benign. Future studies should include comprehensive safety assessments, particularly for materials incorporating nanoparticles or novel chemical compounds. Additionally, research should explore the potential for biodegradable or eco-friendly PHS-NMS to minimize environmental impact.
According to the reviewer's opinion, the text of the summary part was simplified and organized.
Future research should prioritize the development of multifunctional materials with exceptional photothermal and superhydrophobic properties, robust stability, and low cost. Advanced nano-micro manufacturing technologies should be employed to control material microstructures, thereby enhancing light absorption and photothermal conversion efficiencies. Key focuses include improving material stability under low-temperature and high-humidity conditions, optimizing performance under low-intensity solar irradiation (e.g., 0.1 solar radiation intensity), and achieving dual or multiple self-healing functions. Novel preparation technologies, such as structured photothermal energy storage, near-infrared photothermal superhydrophobic materials, and femtosecond laser element doping combined with low-temperature annealing, will drive progress in this field. Additionally, efforts should be directed towards enhancing material durability, exploring cost-effective preparation methods and raw materials, and reducing production costs to accelerate the commercialization of PHS-NMS. In summary, PHS-NMS hold significant application potential and broad development prospects. Future work should focus on systematic research in material design, preparation processes, performance regulation, and optimization to facilitate their practical applications across various fields.
Comments: 15. References: The formatting should follow the Nanomaterials journal guidelines. Currently, the style appears to follow that of Trans. Nonferrous Metals Soc. China.
Response: According to the requirements of the journal, the reference literature format has been modified. Thank you for your careful inspection and important reminder.
Comments: 16. It would be helpful if the authors clarified the criteria used for selecting the references in this article, ensuring that they reflect a comprehensive and up-to-date overview of the field while also providing critical insights into the topic.
Response:
In writing this paper, we have selected references according to the following criteria in order to ensure that they reflect the comprehensiveness and latest developments in the field:
(1) Timeliness of literature:
The authors prioritize research findings published in recent years (the past 5-10 years) to ensure that the cited literature reflects the cutting-edge developments in current research. For example, the references [72]-[102] cited in the text cover studies from 2015 to the present, particularly focusing on the last 3-5 years of research work, to ensure the timeliness of the content.
(2) Authority and representativeness of the literature:
The authors select research findings published in journals with high influence in this field, which are typically rigorously peer-reviewed and possess significant academic value. At the same time, we emphasize citing representative studies to cover the main research directions of carbon-based, polymer-based, semiconductor-based, metal-based, and composite material-based photothermal superhydrophobic materials (PHS-NMS).
(3) Diversity of literature:
To comprehensively reflect the current state of research in this field, we extensively cite various types of studies, including experimental research, theoretical analysis, and applied development. These documents cover a wide range from the preparation of basic materials to practical applications, ensuring that this article provides readers with a comprehensive perspective.
(4) Geographic and diversity of research groups in the literature:
The authors tried to select literature from different countries and research teams to avoid the limitations of a single research group. This diversity helps to illustrate the global research dynamics in this field.
In the study of photothermal (super)hydrophobic materials (PHS-NMS), the authors note the following points that require critical thinking:
(1) Stability of material properties: Despite numerous studies demonstrating the excellent performance of photothermal superhydrophobic materials under laboratory conditions, their long-term stability remains a challenge in practical applications. For instance, carbon-based materials exhibit poor structural stability in complex environments and are susceptible to oxidation and mechanical wear. Moreover, the thermal stability and photothermal conversion efficiency of polymer-based materials need further improvement.
(2) Cost and benefit balance: Metal-based and semiconductor-based photothermal superhydrophobic materials, though excellent in performance, are costly, limiting their large-scale application. In contrast, composite material-based photothermal superhydrophobic materials achieve a better balance between performance and cost by combining the advantages of multiple materials, but their preparation processes are complex, and compatibility issues still need to be addressed.
(3) Limitations in practical applications: At present, most research focuses on the testing of material properties under laboratory conditions, while the impact of actual application environments (such as extreme climate and mechanical vibration) on material properties has not been fully studied. For example, in outdoor environments, the photothermal properties of materials may be significantly affected by light intensity, angle, and ambient temperature.
(4) Environmental adaptability and sustainability: When selecting materials, environmental adaptability and sustainability are indispensable factors. Many high-performance materials rely on rare or toxic chemicals, which can pose potential environmental hazards. Therefore, developing green and sustainable photothermal superhydrophobic materials is a crucial direction for future research.
In summary, although significant progress has been made in the research of photothermal superhydrophobic materials, many key issues still need to be addressed. This paper aims to provide a comprehensive overview of different types of photothermal (super)hydrophobic materials through a comparative analysis, and to offer references for future research directions.
Thank you again for your thoughtful feedback. We look forward to your continued guidance.
Round 2
Reviewer 2 Report
Comments and Suggestions for Authors
Dear Authors, thank you for your response to the review comments. Although in their response to the review, the authors addressed the raised issues, not all changes have been implemented in the manuscript and some issues still need to be addressed before the manuscript can be considered in its final form:
- The manuscript lacks a section on research methodology and criteria for selecting the cited articles (please see PRISMA requirements for review papers). Please include this information to ensure transparency of your review.
- Sections should not begin with a figure. Please respect the reader by first introducing explanatory text. In its current form, the structure does not appear polished for a scientific article.
- In response to the review, you have prepared a table for Section 4. However, the most important data should be summarized and incorporated into the manuscript in a more structured way.
- The text, figures and references require further editing and corrections. Please carefully review the formatting and citation style.
- When citing multiple authors, follow the standard format, e.g., Rekuviene et al. [5].
Please revise the manuscript accordingly to ensure proper publication standards are maintained.
Author Response
Dear Reviewer,
Thank you very much for your valuable comments.
Comments 1.: The manuscript lacks a section on research methodology and criteria for selecting the cited articles (please see PRISMA requirements for review papers). Please include this information to ensure transparency of your review.
Response: Thank you for the opportunity to address the reviewer’s comments and enhance the quality of our manuscript. To meet the PRISMA requirements and ensure the transparency of our review process, we have added a dedicated section titled “2. Methods” to the paper. The full text of this added section is provided below:
- Methods
This review aims to systematically summarize the recent advancements in photothermal and hydrophobic surfaces with nano-micro structures (PHS-NMS) for anti-icing applications. The primary objectives are to: (1) elucidate the mechanisms of ice prevention; (2) describe the fabrication methods of PHS-NMS; (3) explore pathways for performance optimization; and (4) evaluate the anti-icing performance of PHS-NMS in various application scenarios.
Studies were eligible for inclusion if they: (1) focused on photothermal and hydrophobic surfaces with nano-micro structures for anti-icing applications; (2) were published in English; (3) were peer-reviewed journal articles; and (4) provided detailed information on material fabrication, performance evaluation, or application case studies. Exclusion criteria included: (1) non-English publications; (2) non-peer-reviewed articles (e.g., conference proceedings, preprints); (3) review articles or book chapters; and (4) studies unrelated to anti-icing applications.
A comprehensive literature search was conducted using the following databases: Web of Science, Scopus, and Google Scholar. The search terms included "photothermal surfaces," "hydrophobic surfaces," "nano-micro structures," "anti-icing," and "fabrication methods." The search was limited to articles published between January 2010 and December 2024. The search was conducted in January 2025.
The selection process involved two stages: (1) initial screening based on titles and abstracts to identify potentially relevant studies; and (2) full-text evaluation to assess eligibility according to the inclusion and exclusion criteria. Two reviewers independently conducted the screening and selection process. Discrepancies were resolved through discussion and consensus.
Data extracted from eligible studies included: (1) study objectives and background; (2) types of materials and nano-micro structures used; (3) fabrication methods; (4) performance evaluation metrics (e.g., photothermal conversion efficiency, hydrophobicity, anti-icing performance); and (5) application scenarios. The extracted data were organized into tables and discussed in the context of the review objectives.
The quality of the included studies was assessed based on the clarity of research objectives, appropriateness of methods, validity of results, and relevance to anti-icing applications. Studies with detailed experimental procedures, clear performance metrics, and relevant application case studies were considered high quality.
Comments 2.: Sections should not begin with a figure. Please respect the reader by first introducing explanatory text. In its current form, the structure does not appear polished for a scientific article.
Response: We sincerely appreciate your attention to detail and understand the importance of adhering to scientific writing conventions. In response to your comment, we have revised the manuscript to ensure that no section begins with a figure. We have carefully reviewed the structure of the manuscript and made the following adjustments:
(1) Text Before Figures. Figures 2, figure 3, figure 6 and figure 8 are now strategically placed within the text to support the discussion rather than leading it.
(2) Enhanced Readability. The revised manuscript now includes transitional sentences and headings to guide the reader through the content. This approach ensures that the structure appears polished and adheres to the conventions of scientific writing.
(3) Consistent Formatting. We have also reviewed the overall formatting of the manuscript to ensure consistency and clarity. This includes the placement of captions, legends, and references, all of which are now aligned with standard scientific article formatting.
Comments 3.: In response to the review, you have prepared a table for Section 4. However, the most important data should be summarized and incorporated into the manuscript in a more structured way.
Response: Because of the addition of “Section 2. Methods”, the original manuscript's section 4 was changed to section 5 in the revised version. To address the reviewer's comment regarding the need for a more structured summary of the most important data in Section 5, we have revised the section to include a concise and organized summary of key findings and data from each subsection. This will enhance the clarity and readability of the manuscript. The revised contents are listed below.
5.1. Carbon-based PHS-NMS
These studies represent the state-of-the-art in carbon-based PHS-NMS, achieving remarkable photothermal efficiency and hydrophobicity. They effectively address critical challenges such as maintaining performance at low temperatures and enabling rapid ice removal. However, existing fabrication methods often involve complex processes and rely on expensive materials, which hinder scalability and widespread application. Future research should focus on simplifying fabrication techniques and enhancing material durability to improve practical applicability and commercial viability.
5.2. Polymer-based PHS-NMS
The studies showcase the potential of polymer-based materials to achieve high photothermal efficiency and hydrophobicity through simple and scalable fabrication methods. However, these materials often suffer from relatively low photothermal conversion efficiency and limited thermal stability compared to other material systems. Future developments should focus on improving the photothermal performance and durability of polymer-based PHS-NMS while maintaining their ease of processing and tunability. This will require innovations in material design and fabrication techniques to enhance their practical applicability and commercial potential.
5.3. Semiconductor-based PHS-NMS
These studies exemplify the high photothermal efficiency and tunable hydrophobicity achievable with semiconductor materials, especially through nano-micro structural design. However, the performance of these materials is highly dependent on interband absorption processes and thermal relaxation kinetics, which can be challenging to optimize. Future developments should focus on enhancing the stability and scalability of semiconductor-based PHS-NMS. This includes exploring new materials with broader absorption spectra and improving fabrication methods to achieve uniform nano-micro structures, thereby maximizing photothermal conversion efficiency and hydrophobicity for practical anti-icing applications.
5.4. Metal-based PHS-NMS
The studies highlight the superior photothermal performance and rapid ice removal capabilities of metal-based PHS-NMS. However, these materials often face challenges such as high cost, susceptibility to oxidation, and complex fabrication processes. Future developments should focus on addressing these issues by exploring cost-effective materials, optimizing LSPR effects, and simplifying fabrication methods to enhance durability and scalability. This will be crucial for advancing metal-based PHS-NMS towards practical and widespread anti-icing applications.
5.5. Composite-based PHS-NMS
These studies were highlighted for their ability to demonstrate synergistic effects achieved through composite design, thereby addressing the limitations of single-material systems. However, composite PHS-NMS face challenges such as material compatibility and complex fabrication processes. Future developments should focus on optimizing the interactions between different materials, simplifying composite fabrication methods, and improving scalability without compromising performance. This will be essential for realizing the full potential of composite-based PHS-NMS in practical anti-icing applications.
Comments 4.: The text, figures and references require further editing and corrections. Please carefully review the formatting and citation style.
Response: We fully understand the importance of consistency and accuracy in formatting, figures, and citation styles, and we appreciate your attention to these details. In response to your comments, we have taken the following steps to carefully review and correct these aspects of our manuscript:
(1) Formatting Consistency. We have thoroughly reviewed the entire manuscript to ensure that the text adheres to the required formatting guidelines. This includes consistent use of headings, subheadings, font styles, and line spacing throughout the document.
(2) Figure Quality and Consistency. Each figure has been carefully examined for clarity, resolution, and consistency in presentation. We have ensured that all figures are properly labeled, with clear captions and annotations. Additionally, we have checked the consistency of figure numbering and references within the text to avoid any discrepancies.
(3) Citation Style and Accuracy. We have meticulously reviewed all references to ensure they are formatted according to the specified citation style. This includes checking the consistency of in-text citations, reference list formatting, and ensuring that all cited works are accurately represented. We have also double-checked the accuracy of publication details (e.g., authors, journal names, years, and page numbers) to avoid any errors.
Comments 5: When citing multiple authors, follow the standard format, e.g., Rekuviene et al. [5].
Response: Thank you for your specific guidance regarding the citation format. We have carefully reviewed and revised all citations in our manuscript to ensure they strictly follow the standard format required by the Nanomaterials journal. At present, the journal requires that if the total number of authors is less than 10, the names of all authors should be listed. We appreciate your attention to detail, and we are confident that these revisions meet the journal’s formatting standards. If you have any further suggestions or additional areas for improvement, please do not hesitate to let us know.
Thank you again for your thoughtful feedback. We look forward to your continued guidance.
Round 3
Reviewer 2 Report
Comments and Suggestions for Authors
No comments